# Testing a maximum evaporation theory over saturated land: Implications for potential evaporation estimation

Zhuoyi Tu[1], Yuting Yang[1], Michael L. Roderick[2]

5  [1]State Key Laboratory of Hydroscience and Engineering, Department of Hydraulic Engineering, Tsinghua University, Beijing, China

[2]Research School of Earth Sciences, Australian National University, Canberra, ACT, Australia

*Correspondence to*: Yuting Yang (yuting_yang@tsinghua.edu.cn)

**Abstract.** State-of-the-art evaporation models usually assume the net radiation ($R_n$) and surface temperature ($T_s$; or near-surface air temperature) to be independent forcings on evaporation. However, $R_n$ depends directly on $T_s$ via outgoing longwave radiation and this creates a physical coupling between $R_n$ and $T_s$ that extends to evaporation. In this study, we test a maximum evaporation theory originally developed for global ocean over saturated land surfaces, which explicitly acknowledges the interactions between radiation, $T_s$ and evaporation. Similar to the ocean surface, we find a maximum evaporation ($LE_{max}$) emerges over saturated land that represents a generic trade-off between a lower $R_n$ and a higher evaporation fraction as $T_s$ increases. Compared with flux site observations at the daily scale, we show that $LE_{max}$ corresponds well to observed evaporation under non-water-limited conditions and that the $T_s$ at which $LE_{max}$ occurs also corresponds with the observed $T_s$. Our results suggest that saturated land surfaces behave essentially the same as ocean surfaces at time scales longer than a day and further imply that the maximum evaporation concept is a natural attribute of saturated land surfaces, which can be the basis of a new approach to estimating evaporation.

## 1 Introduction

Potential evaporation ($E_P$), defined as the rate of evaporation ($E$) that would occur under non-water-stressed conditions, determines the upper boundary of $E$ over a specific land surface for a given meteorological forcing. Although $E_P$ is more of a hypothetical variable and is generally very difficult to observe, it is often the starting point for partitioning rainfall between $E$, runoff, and soil moisture changes in hydrological, agricultural, ecological and other related studies (Maes et al., 2019; Milly and Dunne, 2016; Scheff and Frierson, 2014; Schellekens et al., 2017; Sheffield et al., 2012; Vicente-Serrano et al., 2013; Wang and Dickinson, 2012). Over the years, numerous mathematical models have been proposed with varying structures and complexities to quantify $E_P$ (e.g., Allen et al., 1998; Priestley and Taylor, 1972; Penman, 1948; Shuttleworth, 1993; Thornthwaite, 1948). Among them, the Penman-Monteith type models (e.g., either the Open Water Penman model (Shuttleworth, 1993) or the Food and Agriculture Organization Penman-Monteith model (Allen et al., 1998)) are most widely used, given their explicit consideration of the radiative and aerodynamic components of $E$, and are hence generally considered as a physical-based and accurate approximation of the real $E$ processes.

Nevertheless, recent empirical evidence shows that the Penman-Monteith type models perform unsatisfactorily in estimating $E_P$ compared with eddy-covariance observations (i.e., the observed $E$ under non-water-stressed conditions; Maes et al., 2019). Instead, the energy balance-based approaches work better in reproducing $E_P$ in both observations (Maes et al., 2019) and climate model simulations (Milly and Dunne, 2016). From an energy balance point of view, the magnitude of $E$ (or in its energy form – latent heat flux or $LE$) is determined by the energy balance equation,

$$LE = (R_n - G)\frac{1}{1+\beta} \tag{1}$$

with $R_n$ the net radiation (W m$^{-2}$) and $G$ the ground heat flux (W m$^{-2}$), which is often negligibly small over land for time scales longer than a day. In Eq. (1), $\beta$ is the Bowen ratio and represents the ratio of sensible heat flux ($H$) over $LE$ (Bowen, 1926). As a result, $LE$ is determined by the available energy at the evaporating surface (i.e., $R_n - G$) and the ability of that evaporating surface to convert the available energy into $LE$, which is represented by the $1/(1+\beta)$ term and often known as the evaporative fraction. With no restriction on water supply, $\beta$ is known to be a decreasing function of temperature at the evaporating surface ($T_s$) (Aminzadeh et al., 2016; Andreas et al., 2013; Guo et al., 2015; Philip, 1987; Priestley and Taylor, 1972; Slatyer and McIlroy, 1961; Yang and Roderick, 2019). This implies that when water is not limiting, both $T_s$ and the available energy determine the rate of $E$. Hence, with fixed available energy, a higher $T_s$ corresponds to a lower $\beta$ (or a higher evaporative fraction) and therefore a larger $LE$. This line of reasoning has directly led to the development of energy balance-based evaporation models, including the classic Equilibrium evaporation approach (Slatyer and McIlroy, 1961) and the Priestley-Taylor evaporation model (Priestley and Taylor, 1972). Compared with Penman-Monteith type models, the energy balance-based approach simplifies the representation of the aerodynamic component of $E$ and usually takes the aerodynamic component of $E$ as a fixed fraction of its radiative counterpart (e.g., 0.26 in the Priestley-Taylor model).

However, a key issue in the above energy balance-based approach is that it takes $R_n$ to be an independent forcing of $E$. A similar idea was also adopted in Penman-Monteith type models (Penman, 1948; Monteith, 1965). Nevertheless, it is clear that $R_n$ cannot be physically independent of either $E$ or $T_s$. On one hand, a higher $T_s$ corresponds to a higher outgoing longwave radiation and therefore a lower

$R_n$. On the other hand, a higher $E$ is associated with a larger evaporative cooling, which lowers $T_s$ and ultimately feedbacks to $R_n$. This latter process confirms that $T_s$ is not independent of $E$. Consequently, the intrinsic interdependence between $R_n$, $E$ and $T_s$ has long been ignored in the state-of-art evaporation models that require $R_n$ as model input (Yang and Roderick, 2019).

To deal with the above issue, a recent study by Yang and Roderick (2019) explicitly considered the interdependence between radiation, $T_s$ and evaporation and tested the new approach over global ocean surfaces. They found that with the increase of $T_s$, $R_n$ decreases while evaporative fraction increases (since $\beta$ decreases as $T_s$ increases) in agreement with a number of previous studies (Aminzadeh et al., 2016; Andreas et al., 2013; Guo et al., 2015; Philip, 1987; Priestley and Taylor, 1972; Slatyer and McIlroy, 1961). This generic and explicit trade-off between a lower $R_n$ and a higher evaporative fraction with the increase of $T_s$ directly yields a maximum evaporation along the $T_s$ gradient according to Eq. (1) (Yang and Roderick, 2019, also see Sect. 2.2). This maximum evaporation emerges naturally from the $R_n$-$T_s$-$E$ interactions and does not require a priori knowledge of $T_s$ thereby alleviating the need for the assumption that $R_n$ and $T_s$ are independent of $E$ in traditional evaporation models. As a result, the maximum evaporation theory does not consider $R_n$ to be an independent forcing of $E$. Instead, it only requires the incoming and reflected solar radiation and an assumption that $\beta$ decreases with the increase of $T_s$ (see Sect. 2.2). Compared with observations of ocean surface evaporation and temperature, Yang and Roderick (2019) demonstrated the validity of the maximum evaporation theory over global ocean surfaces. Here, we test this new maximum evaporation theory over land by asking and answering two questions: does the theory recover the (i) observed $E$ and (ii) does it recover the observed $T_s$? By recovering, we mean that the maximum $E$ as per theory corresponds to the observed $E$ and the $T_s$ at which the maximum $E$ occurs corresponds to the observed $T_s$ under non-water-stressed conditions. Testing the maximum evaporation theory over land is important, as vegetation transpiration generally dominants the total evaporative flux over land (Jasechko et al., 2013; Lian et al., 2018), which is essentially different from ocean surfaces where the evaporative flux only consists of evaporation from open water surfaces. In addition, land surfaces usually have a larger surface roughness than ocean surfaces, which may result in a different energy partitioning (into sensible heat and latent heat) between the ocean and the land. Therefore, it is crucial to test the maximum evaporation theory over land to

determine whether saturated land behaves like the ocean surface and whether the maximum evaporation theory can be the basis of a new approach to estimating $E_P$ over land.

## 2 Materials and methods

### 2.1 Flux site observations

Observations of daily actual evaporation (or latent heat flux), sensible heat flux, ground heat flux along with relevant meteorological variables, radiative fluxes and soil moisture were originally obtained from 212 flux sites collected in the FLUXNET2015 database (http://fluxnet.fluxdata.org/data/fluxnet2015-dataset/). Only days with the data quality metric for $LE$ and $H$ higher than 0.9 (on a scale of 0-1) were used. The daily scale variables were obtained based on 15-min/30min observations using the standard approach (Pastorello et al., 2015). The residual approach (i.e., assuming the observed $H$ is correct and $LE$ is considered as the residual of the energy balance equation) was used to recalculate the fluxes based on a forced energy balance closure at each flux site (Ershadi et al., 2014). We also used the Bowen ratio approach (Twine et al., 2000) to force the flux-site energy balance closure and this resulted in similar model performance (Supplementary Figure S1). Surface temperature for each site-day combination was calculated based on the observed longwave radiation following:

$$T_s = \sqrt[4]{\frac{R_{lo} - (1-\varepsilon)R_{li}}{\varepsilon\sigma}} \tag{2}$$

where $R_{lo}$ and $R_{li}$ are respectively the outgoing and incoming longwave radiation, $\sigma$ is the Stefan-Boltzmann constant ($5.67 \times 10^{-8}$ W m$^{-2}$ K$^{-4}$) and $\varepsilon$ is the surface emissivity, which is acquired from the MODIS (Moderate Resolution Imaging Spectroradiometer) emissivity product (i.e., MOD11A1 Version 6; https://lpdaac.usgs.gov/products/mod11a1v006). The MOD11A1 surface emissivity has a daily temporal resolution and a 1 km spatial resolution. To obtain the emissivity for each EC flux site, we center on the pixel where the site is located and take the mean value of the 81 neighbouring pixels (9×9 pixels) as the emissivity value of the site. For conditions when the MOD11A1 emissivity are not available, we deleted these site-days.

To select a subset of observations at each flux site in which the actual evaporation is not limited by water availability, the energy balance criterion and the soil moisture criterion used by Maes et al (2019)

were adopted. Specifically, at each flux site, the evaporative fraction $EF$ (i.e., $EF=LE/(LE+H)$) was first calculated and the unstressed measurements consisted of all days with $EF$ exceeding the 95[th] percentile $EF$ threshold at each site. Following that, we removed days with soil moisture (averaged over all measured depths) lower than 50% of the maximum soil moisture (taken to be the soil moisture at the 98[th] percentile) at each site. In addition, any remaining site-days with daily $EF$ lower than 0.6 were also removed. Finally, we removed days having a negative $H$ value (account for ~5% of the total daily data) to avoid dealing with strongly advective conditions when accurate measurements are not guaranteed (Paw et al., 2000; Wilson et al., 2002). As a result, a total of 1128 non-water-stressed site-days from 86 sites passed the above criterion and were used in this study (Figure 1 and Supplementary Table S1).

## 2.2 The maximum evaporation model

### 2.2.1 Overview of the maximum evaporation model

The maximum evaporation model calculates evaporation from a wet surface based essentially on surface energy balance (Eq. (1)) with $R_n$ and $\beta$ both explicitly represented as functions of $T_s$ (Yang and Roderick, 2019):

$$LE = \frac{1}{1+\beta(T_s)}[R_n(T_s)-G] \tag{3}$$

In the above equation, the first term on the right-hand side (i.e., $1/[1+\beta(T_s)]$) is the evaporative fraction, which is the ratio of the latent heat flux over the total available energy. Over wet surfaces, since the Bowen ratio decreases with $T_s$ (Aminzadeh et al., 2016; Andreas et al., 2013; Guo et al., 2015; Philip, 1987; Priestley and Taylor, 1972; Slatyer and McIlroy, 1961; Yang and Roderick, 2019), evaporative fraction increases with $T_s$. On the other hand, the second term on the right-hand side of Eq. (3) is the total available energy, which decreases with the increase of $T_s$ as a higher $T_s$ directly leads to a higher outgoing longwave radiation and hence a lower $R_n$ (Yang and Roderick, 2019). As a result, the trade-off between a higher evaporative fraction and a lower $R_n$ with the increase of $T_s$ would naturally lead to a maximum $LE$ along the $T_s$ gradient according to Eq. (3). A previous study by Yang and Roderick (2019) have demonstrated that this naturally emergent maximum $LE$ corresponds well to the actual $LE$ over global ocean surfaces and the $T_s$ at which the maximum $LE$ occurs also corresponds to the observed sea

surface temperature. Here we will test whether this maximum evaporation approach is also valid over
land under non-water-stressed conditions.

### 2.2.2 Parameterization of $R_n$ and $\beta$ as a function of $T_s$

To explicitly acknowledge the dependence of $R_n$ on $T_s$, $R_n(T_s)$ is expressed as:

$$R_n(T_s) = R_{sn} + \varepsilon\sigma(T_s - \Delta T)^4 - \varepsilon\sigma T_s^4 \tag{4}$$

where $R_{sn}$ is the net shortwave radiation (W m$^{-2}$) and is taken to be unchanged with $T_s$. $\Delta T$ is the
temperature difference between $T_s$ and the effective radiating temperature of the atmosphere ($T_{rad}$;
assuming blackbody radiation, $T_{rad}=\sqrt[4]{R_{li}/\sigma}$) and is parameterized as a function of atmospheric
transmissivity and geographic latitude (Yang and Roderick, 2019),

$$\Delta T = n_1 \exp(n_2\tau) + n_3 |lat| \tag{5}$$

where $\tau$ is the atmospheric transmissivity for shortwave radiation (dimensionless) and is calculated as
the ratio of incoming shortwave radiation at the Earth's surface to that at the top of the atmosphere. The
parameter $lat$ is the geographic latitude (in decimal degrees), which is considered here to account for a
longer pathway of short-wave radiation going through the atmosphere in higher latitudes compared to
lower latitudes. $n_1$, $n_2$, and $n_3$ are fitting coefficients. Using extensive data over the global ocean ($n$ =
202,794), Yang and Roderick (2019) determined the values of these coefficients to be $n_1$=2.52, $n_2$=2.38
and $n_3$=0.035, respectively. Here, we directly adopt these same coefficient values over land for two
reasons: (i) the key processes governing the interactions between incoming and outgoing longwave
radiations are essentially the same for ocean and land (mainly greenhouse gas and aerosol effects), and
(ii) there were many more samples available for parameterizing Eq. (5) over the ocean than that over
land. Validation against observations from all 1128 non-water-limited site-days demonstrates an overall
good performance of Eq. (5) in estimating $\Delta T$ over land under saturated conditions (Supplementary
Figure S2).

The Bowen ratio ($\beta$) is expressed as a function of $T_s$:

$$\beta(T_s) = m\frac{\gamma(T_s)}{\Delta(T_s)} \tag{6}$$

where $m$ is a fitting coefficient. $\gamma$ is the psychrometric constant (kPa K$^{-1}$), and $\Delta$ is the slope of the saturation vapor pressure curve (kPa K$^{-1}$), both of which are functions of $T_s$:

$$\gamma(T_s) = \frac{C_P P_a}{0.622 L(T_s)} \tag{7}$$

$$\Delta(T_s) = \frac{4098 e_s(T_s)}{(T_s - 35.8)^2} \tag{8}$$

where $C_P$ is the specific heat of air at constant pressure (1.01 kJ kg$^{-1}$ K$^{-1}$), $P_a$ is the air pressure (kPa), $e_s$ is the saturated vapor pressure (kPa). $L$ is the latent heat of vaporization (kJ kg$^{-1}$) and is calculated as weak function of temperature:

$$L(T_s) = 2.51 \times 10^3 - 2.32 \times (T_s - 273.15) \tag{9}$$

To apply the maximum evaporation model, an array of $T_s$ (e.g., from 250 K to 330 K at an interval of 0.1 K) is generated along with the observed $R_{sn}$ and $G$ and these are applied to Eq. (4) and Eq. (6) and then Eq. (3) to estimate $LE$ at each corresponding $T_s$. The maximum evaporation is then located in that array as well as the surface temperature at which this maximum occurs (see Figure 3 for an example).

## 3 Results

The maximum evaporation theory is tested at 86 flux sites globally, covering a wide range of bio-climates (Figure 1 and Supplementary Table S1). By pooling daily observations of $H$, $LE$ and $T_s$ across all 1128 site-days, we first obtain a generic $\beta$-$T_s$ relationship as $\beta = 0.27\gamma/\Delta$. Similarly, we also obtained a $\beta$-$T_s$ relationship for each separate biome type as shown in Figure 2. By comparison, Yang and Roderick (2019) reported a $\beta$-$T_s$ relationship over ocean as $\beta = 0.24\gamma/\Delta$. This means that for the same $T_s$, $\beta$ over land is generally larger than that over ocean. Interestingly, the ocean surface $\beta$-$T_s$ relationship is identical to that in wetlands obtained here. These $\beta$-$T_s$ relationships will be used in the following calculations of $LE$ using the maximum evaporation approach.

To get an overview of how each of the energy fluxes varies with $T_s$ we first examine the maximum evaporation theory using the pooled data over all 1128 site-days (Figure 3). Under this condition, the mean observed net shortwave radiation ($R_{sn}$) over all site-days is 176.6 W m$^{-2}$ and $G$ is 1.0 W m$^{-2}$. Since

$R_{sn}$ is not directly dependent on $T_s$ and $G$ is negligibly small, the term $R_{sn}$ minus $G$ is held constant across the entire $T_s$ range. With the increase of $T_s$, it is readily apparent that both outgoing and incoming longwave radiation ($R_{lo}$ and $R_{li}$) steadily increase (see Sect. 2.2 for details about the coupling between $R_{lo}$ and $R_{li}$), with $R_{lo}$ increasing slightly faster than $R_{li}$, leading to a decreased net longwave radiation and thus a decreased $R_n$ as $T_s$ increases (Figure 3). With this and the observed generic dependence of $\beta$ on $T_s$ ($\beta = 0.27\gamma/\Delta$, Figure 2), a maximum $LE$ emerges along the $T_s$ gradient that represents the interaction between decreasing $R_n$ and increasing evaporative fraction as $T_s$ increases. For the pooled dataset used here, the maximum $LE$ ($LE_{max}$) is found to be 105.6 W m$^{-2}$ and the corresponding $T_s$ is 294.7 K, both of which are very close to the averages computed from all daily flux site observations (i.e., $LE_{obs} = 102.4$ W m$^{-2}$ and $T_{s\_obs} = 292.3$ K) (Figure 3).

Having demonstrated the overall concept, we next perform the detailed calculations using data for all individual site-days (Figures 4-6). Using the same generic $\beta$ dependence on $T_s$ ($\beta = 0.27\gamma/\Delta$), $LE_{max}$ estimated from the maximum evaporation model agrees very well with flux site observations, yielding an $R^2$ of 0.92, a root-mean-squared error (RMSE) of 14.6 W m$^{-2}$ and a mean bias of 1.6 W m$^{-2}$ (Figure 4a). The performance of the maximum evaporation model improves slightly when the biome-specific model parameters are used (RMSE decreases to 14.1 W m$^{-2}$ and mean bias decreases to 1.4 W m$^{-2}$; Figure 4b). This result demonstrates that $LE_{max}$ corresponds to the observed evaporation under well-watered conditions across a broad range of bio-climates. In fact, when the previously identified ocean surface $\beta$-$T_s$ relationship is adopted, the maximum evaporation approach performs only slightly worse than those based on the calibrated $\beta$-$T_s$ relationship over saturated lands, yielding an $R^2$ of 0.91, an RMSE of 14.8 W m$^{-2}$ and a mean bias of 2.8 W m$^{-2}$ (Figure 4c).

We next test whether the maximum evaporation approach could recover $T_s$ over the same saturated land surfaces. Similar to the test of $LE$, the three $\beta$-$T_s$ relationships are respectively used. Results show that when the generic $\beta$-$T_s$ relationship over land is used (i.e., $\beta = 0.27\gamma/\Delta$), the $T_s$ at which $LE_{max}$ occurs corresponds reasonably well to the observed $T_s$, with an $R^2$ of 0.62, an RMSE of 4.3 K and a mean bias of 0.3 K (Figure 5a), indicating that the maximum evaporation approach is also able to recover $T_s$ under saturated conditions. Again, the model's performance in recovering $T_s$ increases slightly when the

biome-specific $\beta$-$T_s$ relationships are used (Figure 5b). When the ocean surface $\beta$-$T_s$ relationship is used, the model performs similarly in estimating the variability of $T_s$ to that of the generic land $\beta$-$T_s$ relationship (Figure 5c). However, the ocean surface $\beta$-$T_s$ relationship (Figure 5c) results in a higher $T_s$ mean bias compared to the $\beta$-$T_s$ relationships obtained over land (Figure 5a).

Different from most state-of-the-art evaporation models, the maximum evaporation approach does not rely on observed $R_n$ (or independent $R_n$ estimates) as model input but estimates $R_n$ as a result of the $R_n$-$T_s$-$E$ interaction. Here, we also test the estimated $R_n$ calculated using the maximum evaporation approach as the discrepancy between $LE_{max}$ and $LE_{obs}$ is mainly caused by the slight difference between $T_{s\_max}$ and $T_{s\_obs}$ that leads to different $R_{lo}$ and $R_{li}$ (and thus a different $R_n$) to be used in the calculation of $LE_{max}$. It should be noted that since the observed shortwave radiation is used in the maximum evaporation model, validation of $R_n$ is essentially the same as the validation of net longwave radiation. We find that the maximum evaporation model could satisfactorily reproduce the observed $R_n$ when the generic land $\beta$-$T_s$ relationship is used, as indicated by an $R^2$ of 0.93, an RMSE of 14.4 W m$^{-2}$ and a mean bias of 2.3 W m$^{-2}$ (Figure 6a). Using biome-specific $\beta$-$T_s$ relationships or the ocean surface $\beta$-$T_s$ relationship does not considerably increase or decrease the model's performance in estimating $R_n$ (Figures 6b and 6c).

## 4 Discussion

Taking $R_n$ and/or $T_s$ (or near-surface air temperature) to be independent forcings has long been identified as a scientific concern in the use of evaporation models (Milly, 1991; Monteith and Unsworth, 2013; Philip, 1987). Here, we test a maximum evaporation theory developed over the global ocean surface that addresses this concern by explicitly acknowledging the interdependence between radiation, surface temperature and evaporation (Yang and Roderick, 2019). Our new results show that there exists a maximum evaporation along the $T_s$ gradient that corresponds to the observed evaporation under saturated conditions over land (Figures 3 and 4). In addition, the $T_s$ at which $LE_{max}$ occurs also corresponds reasonably well to the observed $T_s$ (Figures 3 and 5). These results mirror those found previously over the global ocean (Yang and Roderick, 2019). This is not a surprise since the basic

principles are the same for a wet land surface and the ocean surface. These results suggest that saturated land surfaces behave essentially the same as ocean surfaces and imply that $LE_{max}$ is a natural attribute of the land surface when water availability does not limit evaporation.

A key assumption involved in the maximum evaporation model is that $\beta$ decreases with the increase of $T_s$ under saturated conditions. Nevertheless, this key assumption that $\beta$ decreases with the increase of $T_s$ under saturated conditions has been extensively validated in previous studies based on theoretical relationships (Philip, 1987; Priestley and Taylor, 1972; Slatyer and McIlroy, 1961; Lhomme, 1997) and *in situ* observations (Andreas et al., 2013; Guo et al., 2015; Yang and Roderick, 2019; also see Supplementary Figure S3). Moreover, our results also found this held over saturated lands (Figure 2). The original maximum evaporation study reported that $\beta = 0.24\gamma/\Delta$ over global ocean surfaces (Yang and Roderick, 2019). Here, we find the generic land surface coefficient increases to 0.27 (i.e., $\beta = 0.27\gamma/\Delta$, Figure 2) which indicates a slightly higher $\beta$ over wet vegetated land than that over the ocean surface for the same $T_s$. This is biophysically reasonable, as the stoma of plant leaves represents an additional resistance to vapor transfer between the land and the atmosphere (Swann et al., 2016; Yang et al., 2019), which lowers the ability of a generic vegetated surface to convert available energy into $LE$ for a given $T_s$, compared to open water surfaces. In addition, different surface roughness can also lead to different $\beta$-$T_s$ relationships between the land and the ocean. Compared with the ocean surface that shows a tight $\beta$-$T_s$ relationship (Yang and Roderick, 2019), the $\beta$-$T_s$ relationships over saturated vegetated land are relatively weak with considerable scatter (Figure 2). This data scatter could be caused by several reasons. First, the observations by eddy covariance (EC) towers can be a source of uncertainty. This is threefold, including (i) the quality of the observations, (ii) the footprint within each EC tower may be heterogeneous (Lee et al., 2004; Paw et al., 2000), and (iii) whether the selected days are truly non-water-limited (however, see Supplementary Figure S4). Second, as is seen in Figure 2, different biome types exhibit different $\beta$-$T_s$ relationships. This can be caused by different surface resistance and roughness between biome types and even between sites. Nevertheless, these data-based limitations only have limited impacts on the model performance, as similar performance is obtained using both the generic $\beta$-$T_s$ relationship (i.e., $\beta = 0.27\gamma/\Delta$) and biome-specific $\beta$-$T_s$ relationships (Figure 4). Third, wind speed could be another factor that leads to the scatter. For the same surface roughness, a

different wind speed will lead to a different aerodynamic resistance and therefore a different $\beta$. However, this effect is usually very small, as demonstrated by the long-standing similarity theory (the transfer of mass and heat share the same aerodynamic process in the lower atmospheric boundary layer; Monin and Obukhov, 1954). In fact, our findings that one can make a reasonable estimate of $LE$ using a generic land or ocean $\beta$-$T_s$ relationship instead of a site-specific relationship (Figure 4) imply that $R_n$ is the primary determinant of $LE$ over saturated surfaces. As evaporation tends to operate at its maximum strength, sensible heat (and $\beta$) are usually very small over warm saturated land surfaces. As a result, once $R_n$ can be accurately determined, any reasonable $\beta$-$T_s$ relationship (Figure 2) would result in a satisfactory $LE$ estimate (Figure 4 and Supplementary Figure S5). Our result highlighted in Figure 3 shows that $R_n$ (and hence $LE$) is only a weak function of $T_s$ and this explains why one can obtain an accurate estimate of $LE$ using a generic $\beta$-$T_s$ relation. However, the same logic also leads to the conclusion that an accurate $\beta$-$T_s$ relationship will be necessary to estimate $T_s$, since $T_s$ is very sensitive to changes in $LE$ (Figure 3). In this regard, using the land $\beta$-$T_s$ relationships (preferably site-specific relations) is preferable to a generic ocean surface relation (Figure 5). To further demonstrate the above points, we conduct an uncertainty test by varying the coefficient $m$ in the $\beta$-$T_s$ relationship in Supplementary Figure S6. We find that when $m$ ranges from 0.18 to 0.36 (all other forcings as per Figure 3), the change in estimated $LE_{max}$ is only 9 W m$^{-2}$ (101.7 – 110.7 W m$^{-2}$), whereas the change in estimated $T_{s\_max}$ is as high as 11.6 K (287.9 – 299.5 K).

Besides the data scattering that leads to an uncertainty in the $\beta$-$T_s$ relationships, there are also uncertainties associated with (i) parameterization of the longwave coupling and (ii) selection of non-water-stressed observations in the current study. In the maximum evaporation approach, the coupling between outgoing and incoming longwave radiation is calculated using the temperature difference between the surface and an effective radiating height in the atmosphere ($\Delta T$) and is parameterized as a function of shortwave atmospheric transmissivity and geographic latitude. However, the shortwave atmospheric transmissivity is primarily affected by aerosols while the longwave transmissivity is mainly affected by the concentration of greenhouse gases. Nevertheless, here we only deal with wet conditions, under which the vapour concentration of the atmosphere is also relatively high and more aerosols would favour the development of more clouds that simultaneously affect both shortwave and longwave

radiations. We suspect that this underlies the excellent performance of Eq. (5) in estimating $\Delta T$ at the flux sites (Supplementary Figure S2). To further evaluate that conclusion, we additionally evaluate the estimated longwave radiation against four global products (i.e., ERA5, Hersbach et al., 2019; CERES,

Kato et al., 2018; the Princeton global forcing data, Sheffield et al., 2006; the GLDAS global forcing data, Rodell et al., 2004) and compare our longwave estimates with other two semi-empirical models (i.e., Brutsaert, 1975 and Shakespeare and Roderick, 2021). The results show our $\Delta T$-based approach to be the best performer across a wide of conditions when the surface is wet (Supplementary Figure S7). In addition, we further note that our maximum evaporation model is only tested at the daily time scale

(Figures 4-6) and longer (Figure 3). In particular, for time scales shorter than that (e.g., hourly), the diurnal cycle of $E$ can be very different for ocean and land surfaces (Kleidon and Renner, 2017). In addition, the parameterization of the coupling between incoming and outgoing longwave radiation via Eq. (5) requires a time scale that is long enough to allow the surface heat fluxes to be fully redistributed through the atmospheric column (Yang and Roderick, 2019). At sub-daily scales, Eq. (5) is likely

invalid because $R_{lo}$ usually exhibits a larger diurnal range than $R_{li}$ during a typical cloudless day (Monteith and Unsworth, 2013).

As for the selection of non-water-stressed evaporation observations from global EC towers, we rely largely on the same selection criteria used in a previous study (Maes et al., 2019). However, these selection criteria are somewhat subjective and represent a compromise between better data quality and

more data samples. As a result, the selected site-days are not necessarily non-water-limited. Nevertheless, varying the selection criteria (changing the thresholds) of non-water-stressed evaporation only resulted in minor changes in the overall model performance (Supplementary Figure S4), which suggests that the uncertainties in the selection of non-water-stressed evaporation observations would not materially change our conclusion.

The ability of the maximum evaporation model to recover $LE$ and $T_s$ over vegetated lands under saturated conditions has an important implication for the estimation of potential evaporation, which is a central concept in hydrology and agriculture (and especially in irrigation). The underlying idea of $E_P$ is straightforward – it is the evaporation that would occur with an unlimited supply of water. However, the

formal physical definition of $E_P$ has been widely debated in the literature (Brutsaert, 2005; Donohue et al., 2010; Granger, 1989; Nash, 1989) and the calculation of $E_P$ using conventional evaporation models is problematic (Aminzadeh et al., 2016; Roderick et al., 2015). The key scientific issue is that the meteorological forcing variables observed over actual surfaces are generally not equivalent to the meteorological variables that would be measured over a hypothetical surface with unlimited water supply. Compared with existing evaporation models, the maximum evaporation model presented here requires fewer meteorological variables than existing approaches (but performs similarly with existing approaches under wet conditions, see Supplementary Figure S8 for details). This new approach only requires the incoming and reflected solar radiation, a relationship that describes a decreasing dependence of $\beta$ on $T_s$, and a relation for the coupling of the incoming and outgoing longwave radiation. With these modest requirements, $LE_{max}$ naturally follows from the physical interdependence between radiation, surface temperature and evaporation. These features suggest that the maximum evaporation model can be used to make a strictly independent estimate of $E_P$. In fact, the maximum evaporation formulation directly maps to one particular definition of $E_p$ that was proposed by Brutsaert (2015) as "the maximum evaporation that would occur over real surfaces with the actual solar forcing and a prescribed Bowen ratio".

## 5 Conclusions

In this study, we test a maximum evaporation theory that explicitly acknowledges the interdependence between radiation, surface temperature and evaporation over saturated land surfaces. Validated against flux site observations, we show that the maximum evaporation approach could recover the observed evaporation across a broad range of bioclimates. In comparison, although the model is also able to reasonably recover the observed $T_s$, the model's performance in recovering $T_s$ is not as good as that for $LE$. Nevertheless, this does not materially lead to larger errors in $LE$ estimates, as we additionally demonstrate that $LE$ is not sensitive to $T_s$ changes. The overall good performance of the maximum evaporation approach over saturated surfaces implies a great potential of the method to be used for estimating potential evaporation. To calculate $E_P$ in practice using the maximum evaporation approach, a detailed site (or biome) specific $\beta$-$T_s$ relationship (e.g., Figure 2) would be favorable; otherwise, a

generic default $\beta$-$T_s$ relationship ($\beta = 0.27\gamma/\Delta$ or even $\beta = 0.24\gamma/\Delta$) can also lead to a reasonable $E_P$ estimate that remains consistent with the $E_P$ definition by Brutsaert (2015). Supplementary Table S2 gives a worked example of applying the maximum evaporation model for $E_P$ estimation.

### Data availability

All data for this paper are properly cited and referred to in the reference list.

### Author contribution

YY and MLR conceived the idea and designed the research. ZT performed the calculation. YY drafted the manuscript. All authors contributed to results discussion and manuscript writing.

### Competing interests

The authors declare that they have no conflict of interest.

### Acknowledgements

This study is financially supported by the Ministry of Science and Technology of China (Grant No. 2019YFC1510604), the National Natural Science Foundation of China (Grant No. 42041004, 42071029, 41890821) and the Guoqiang Institute of Tsinghua University (Grant No. 2019GQG1020). MLR acknowledges the support of the Australian Research Council (DP190100791). The FLUXNET community is greatly appreciated for making the eddy covariance data publicly available. The FLUXNET2015 dataset is acquired from http://fluxnet.fluxdata.org/data/fluxnet2015-dataset/.

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

## List of Figures

**Figure 1.** Location of the 86 flux sites used in this study. Numbers in the brackets indicate the number of sites for each biome type.

**Figure 2.** Relationship between Bowen ratio ($\beta$) and surface temperature ($T_s$) over saturated land surfaces. The thick black curve represents the fitted $\beta$-$T_s$ relationship across all data points (i.e., $n$=1128, $\beta = 0.27\gamma/\Delta$, $R^2$=0.11, $p$<0.001), and the colored lines represent different biome types with the number of data points ($n$ site-days) and fitted $\beta$-$T_s$ relationship for each biome type shown in the legend.

**Figure 3.** Variation of energy fluxes with $T_s$. Plot shows how the energy fluxes vary with $T_s$ for a fixed value of $R_{sn} - G$ at 176.6 W m$^{-2}$ ($R_{sn}$ is the net shortwave radiation, see Eq. (4) in Sect. 2.2). The red dot indicates the maximum evaporation and the red triangle shows the observed evaporation. The $T_s$ at which the maximum evaporation occurs is shown by the dashed vertical line.

**Figure 4.** Comparison of the maximum evaporation and observed evaporation over saturated land surfaces using three different $\beta$-$T_s$ relationships. (a) Generic land $\beta$-$T_s$ relationship ($\beta = 0.27\gamma/\Delta$, $n = 1128$). (b) Biome-specific $\beta$-$T_s$ relationships (per Figure 2). (c) Ocean surface $\beta$-$T_s$ relationship ($\beta = 0.24\gamma/\Delta$, Yang and Roderick, 2019). The colors indicate different biome types (as provided in Figure 1). The dashed black line indicates the 1:1 line.

**Figure 5.** Comparison of the estimated and observed surface temperature over saturated land surfaces using three different $\beta$-$T_s$ relationships. Comparison of estimated surface temperature ($T_{s\_max}$) with flux site observations ($T_{s\_obs}$). (a) Generic land $\beta$-$T_s$ relationship ($\beta = 0.27\gamma/\Delta$, $n = 1128$). (b) Biome-specific $\beta$-$T_s$ relationships (per Figure 2). (c) Ocean surface $\beta$-$T_s$ relationship ($\beta = 0.24\gamma/\Delta$, Yang and Roderick, 2019). The colors indicate different biome types (as provided in Figure 1). The dashed black line indicates the 1:1 line.

**Figure 6.** Comparison of the estimated and observed net radiation over saturated land surfaces using three different $\beta$-$T_s$ relationships. Comparison of estimated net radiation ($R_{n\_max}$) with flux site observations ($R_{n\_obs}$). (a) Generic land $\beta$-$T_s$ relationship ($\beta = 0.27\gamma/\Delta$, $n = 1128$). (b) Biome-specific $\beta$-$T_s$ relationships (per Figure 2). (c) Ocean surface $\beta$-$T_s$ relationship ($\beta = 0.24\gamma/\Delta$, Yang and Roderick, 2019). The colors indicate different biome types (as provided in Figure 1). The dashed black line indicates the 1:1 line.

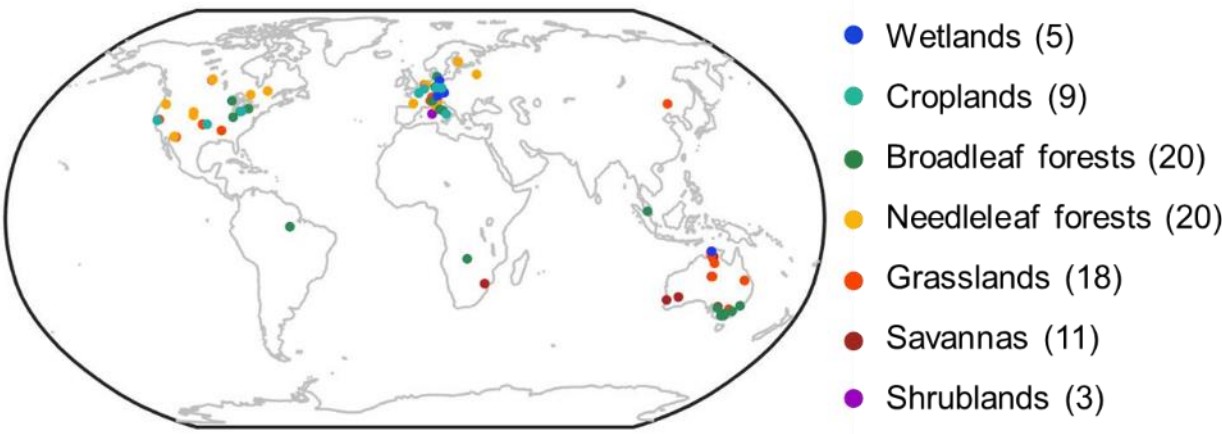

**Figure 1.** Location of the 86 flux sites used in this study. Numbers in the brackets indicate the number of sites for each biome type.

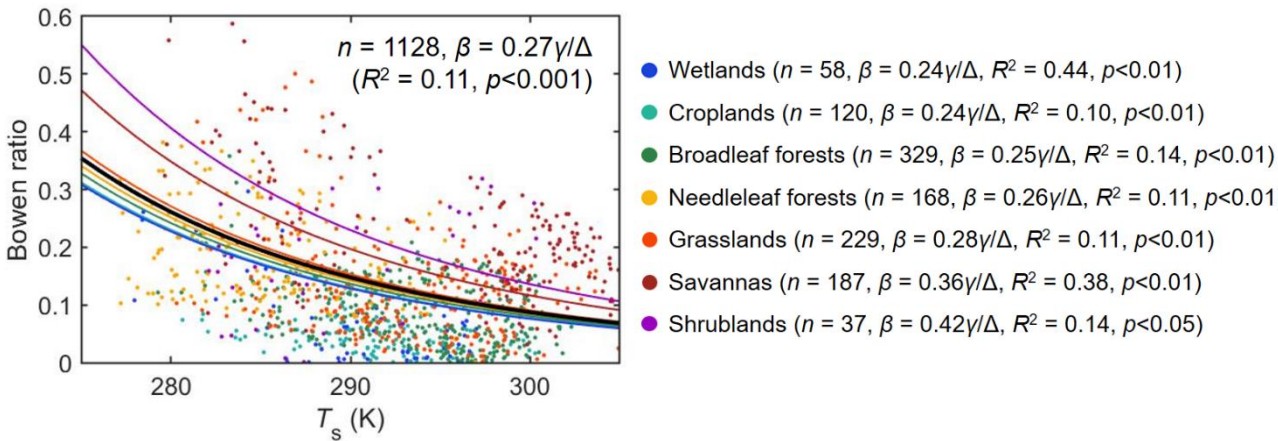

**Figure 2.** Relationship between Bowen ratio ($\beta$) and surface temperature ($T_s$) over saturated land
surfaces. The thick black curve represents the fitted $\beta$-$T_s$ relationship across all data points (i.e., $n=1128$, $\beta = 0.27\gamma/\Delta$, $R^2=0.11$, $p<0.001$), and the colored lines represent different biome types with the number of data points ($n$ site-days) and fitted $\beta$-$T_s$ relationship for each biome type shown in the legend.

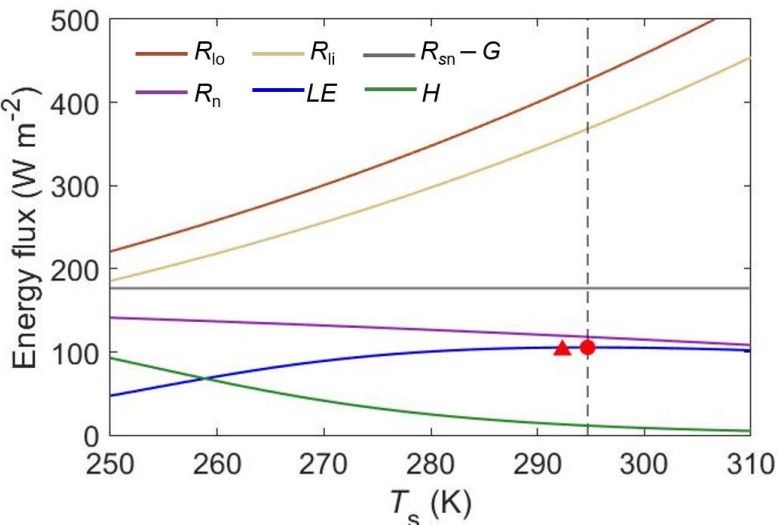

**Figure 3.** Variation of energy fluxes with $T_s$. Plot shows how the energy fluxes vary with $T_s$ for a fixed value of $R_{sn} - G$ at 176.6 W m$^{-2}$ ($R_{sn}$ is the net shortwave radiation, see Eq. (4) in Sect. 2.2). The red dot indicates the maximum evaporation and the red triangle shows the observed evaporation. The $T_s$ at which the maximum evaporation occurs is shown by the dashed vertical line.

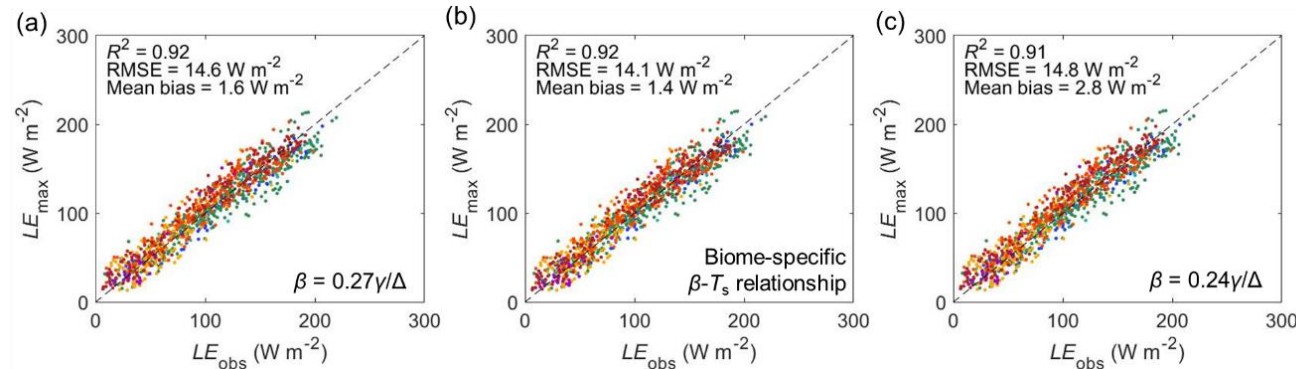

**Figure 4.** Comparison of the maximum evaporation and observed evaporation over saturated land surfaces using three different $\beta$-$T_s$ relationships. (a) Generic land $\beta$-$T_s$ relationship ($\beta = 0.27\gamma/\Delta$, $n = 1128$). (b) Biome-specific $\beta$-$T_s$ relationships (per Figure 2). (c) Ocean surface $\beta$-$T_s$ relationship ($\beta = 0.24\gamma/\Delta$, Yang and Roderick, 2019). The colors indicate different biome types (as provided in Figure 1). 530 The dashed black line indicates the 1:1 line.

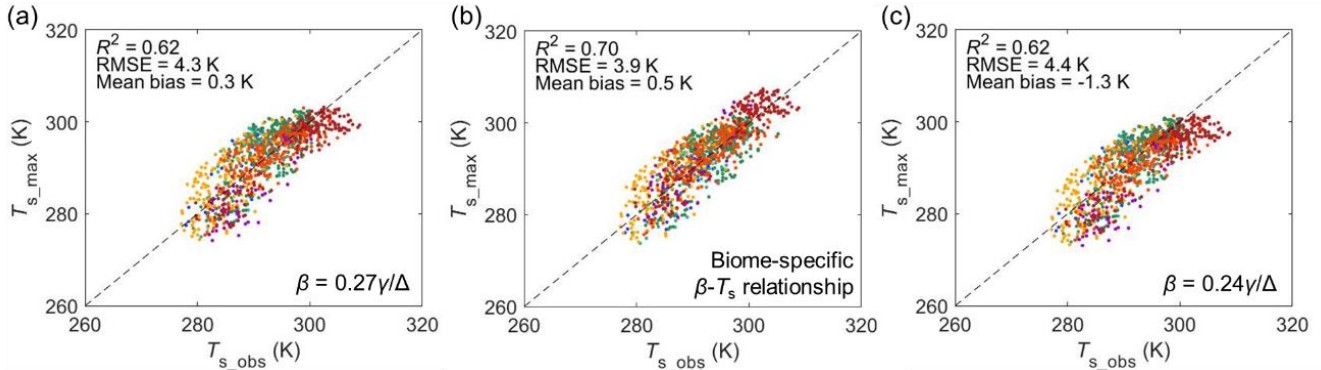

**Figure 5.** Comparison of the estimated and observed surface temperature over saturated land surfaces using three different $\beta$-$T_s$ relationships. Comparison of estimated surface temperature ($T_{s\_max}$) with flux site observations ($T_{s\_obs}$). (a) Generic land $\beta$-$T_s$ relationship ($\beta = 0.27\gamma/\Delta$, $n = 1128$). (b) Biome-specific $\beta$-$T_s$ relationships (per Figure 2). (c) Ocean surface $\beta$-$T_s$ relationship ($\beta = 0.24\gamma/\Delta$, Yang and Roderick, 2019). The colors indicate different biome types (as provided in Figure 1). The dashed black line indicates the 1:1 line.

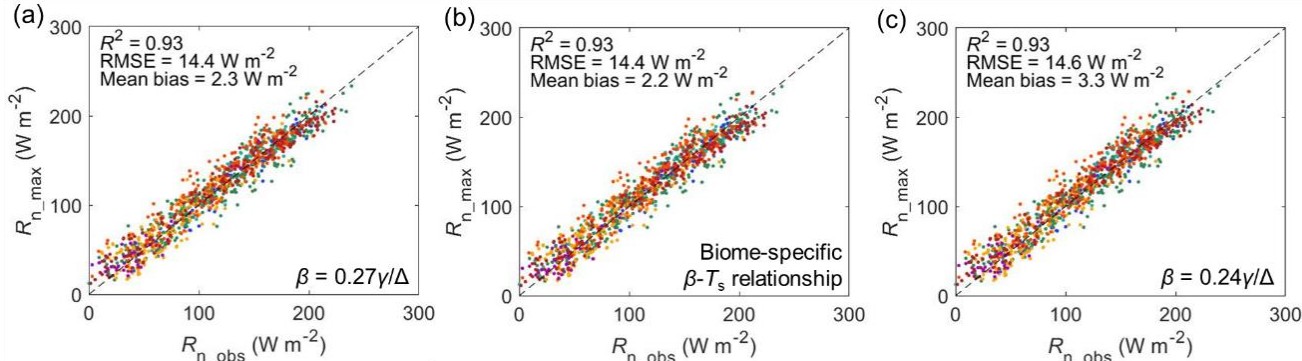

**Figure 6.** Comparison of the estimated and observed net radiation over saturated land surfaces using three different $\beta$-$T_s$ relationships. Comparison of estimated net radiation ($R_{n\_max}$) with flux site observations ($R_{n\_obs}$). (a) Generic land $\beta$-$T_s$ relationship ($\beta = 0.27\gamma/\Delta$, $n = 1128$). (b) Biome-specific $\beta$-$T_s$ relationships (per Figure 2). (c) Ocean surface $\beta$-$T_s$ relationship ($\beta = 0.24\gamma/\Delta$, Yang and Roderick, 2019). The colors indicate different biome types (as provided in Figure 1). The dashed black line indicates the 1:1 line.