# Peer review of "Testing a maximum evaporation theory over saturated land: Implications for potential evaporation estimation"

_Hydrology and Earth System Sciences, 2021_

## Author Response (AR1)

**Response to Reviewers' comments**

We greatly appreciate the anonymous referees for providing valuable and constructive comments that are of great help for us to improve the quality of the manuscript. We have fully considered the comments and the point-to-point responses are listed below. The manuscript and the supplementary material have been revised to accommodate the changes (changes are marked in blue color). In the following the reviewer comments are black font and our responses are blue and to assist with navigation we use codes, such as R1C2 (Reviewer 1 Comment 2)

**To reviewer #1**

**R1C1:** The authors have validated the maximum evaporation theory originally developed for oceans over global saturated land surfaces. I think this paper is a good extension of Yang et al. (2019) and is of great importance for land potential evaporation estimation.

**Response:** Thanks for your positive evaluation and encouraging comments on our manuscript. Your individual comments are replied below.

**R1C2:** In the last paragraph of introduction, the author intended to test their ocean research directly over saturated lands without any comparison between two different surfaces. I suggest to add some discussions on comparison (vegetation effect?) between ocean and land surface, which was mentioned in discussion. I think this kind of comparison can highlight the importance of this research and also can help authors to propose scientific hypothesis.

**Response:** Done. Following your suggestion, We have added a few sentences in the introduction regarding the difference between the ocean and the land (Line 86-93).

**Relevant text reads (Line 86-93):** "Testing the maximum evaporation theory over land is important, as vegetation transpiration generally dominants the total evaporative flux over land (Jasechko et al., 2013; Lian et al., 2018), which is essentially different from ocean surfaces where the evaporative flux only consists of evaporation from open water surfaces. In addition, land surfaces usually have a larger surface roughness than ocean surfaces, which may result in a different energy partitioning (into sensible heat and latent heat) between the ocean and the land. Therefore, it is crucial to test the maximum evaporation theory over land to determine whether saturated land behaves like the ocean surface and whether the maximum

evaporation theory can be the basis of a new approach to estimating $E_P$ over land."

**R1C3:** In introduction, the authors have pointed out the limitations of Penman and Priestley-Taylor model. So please simulating evaporation with this two models at select site-days, and then compare simulations to maximum evaporation method results. To see if maximum evaporation method show higher performance than the two classical models.

**Response:** Done. As suggested, we compared the maximum approach, the Priestley-Taylor model and the open-water-Penman model at the selected site-days (Figure R1). It shows that the maximum evaporation model performs similarly (although slightly worse) with the Priestley-Taylor model, both of which perform evidently better than the Penman model. It is not surprising that the Priestley-Taylor model performs slightly better than the maximum evaporation approach since the Priestley-Taylor model uses the observed net radiation and surface temperature while the maximum approach uses the estimated net radiation and surface temperature (the performance of surface temperature/net radiation estimations are shown in Figure 5 and 6). However, as demonstrated in Yang and Roderick (2019), the underlying interactions between radiation, surface temperature and evaporation in the Priestley-Taylor model are incorrect, which means that the Priestley-Taylor model gets a right answer with a wrong approach. The weakness of the Priestley-Taylor model would not be apparent under wet conditions (as focused here) but would become more evident when the surface becomes drier, since the observed net radiation and surface temperature under dry conditions can be very different from those if the surface were wet (the idea of potential evaporation).

[Figure]

**Figure R1.** Performance of the maximum evaporation approach, the Priestley-Taylor model and the open-water-Penman model in estimating evaporation at selected site-days.

Since the main purpose of this study is to test the maximum evaporation approach over wet lands, we do not plan to include this comparison in the main text. In addition, a previous study by Maes et al. (2019) has already demonstrated that the energy balance-based approaches generally perform better than other approaches (including Penman-Monteith) at flux sites when the surface is wet. In the revised manuscript, we have briefly discussed the difference between the maximum evaporation model and other models and included the comparison results as Figure S8 in the supplementary material (Line 334-336 and Figure S8).

**Relevant text reads (Line 334-336):** "Compared with existing evaporation models, the maximum evaporation model presented here requires fewer meteorological variables than existing approaches (but performs similarly with existing approaches under wet conditions, see Supplementary Figure S8 for details)."

**R1C4:** In section 2.1, the residual approach was used to force energy balance for EC flux data. The method will decrease Bowen ratio because latent heat flux usually increase after adjustment due to lack of energy balance for EC method, while sensible heat keep the same. The residual method not only changed latent heat flux, but also changed the Bowen ratio. And Bowen ratio is a very important variable in your research. I think the residual approach is not the optimal one here. You can try the method proposed by Twine et al. (2000). Twine method assumes that even though the EC latent and sensible heat fluxes are not measured accurately, the resulting Bowen ratio is accurate. Then turbulent fluxes are adjusted without changing the Bowen ratio.

**Response:** Done. Following this comment, we used the Bowen ratio approach noted by the reviewer to close the energy balance and repeated the calculations. We find that using different approaches to close the energy balance results in similar model performance in estimating $LE$ and $T_s$ (Figure R2). This is not surprising, as over saturated surfaces, sensible heat is usually very small. We have added a few sentence in method (Line 103-105) and included Figure R2 as the supplementary material Figure S1 to demonstrate that different approaches to closing the flux site energy balance do not change our conclusion.

**Relevant text reads (Line 103-105):** "We also used the Bowen ratio approach (Twine et al., 2000) to force the flux-site energy balance closure and this resulted in similar model performance (Supplementary Figure S1)."

[Figure]

**Figure R2** Validation of *LE* and $T_s$ estimated using the maximum evaporation approach at the selected site-days where the energy balance closure of the flux site measurements is achieved by using the Bowen ratio approach.

**R1C5:** Equation (2). Please describe obtaining surface emissivity value with MOD11A1 products in more detail, such as time scale (different emissivity value for different day?), spatial scale (the matching between site location and MODIS pixel) and missing data problem (how to deal with conditions with no MOD11A1 for some site-day).

**Response:** Done. The MOD11A1 surface emissivity has a daily temporal resolution and a 1 km spatial resolution. To obtain the emissivity for each EC flux site, we center on the pixel where the site is located and take the mean value of the 81 neighboring pixels (9×9 pixels) as the emissivity value of the site. For conditions when the MOD11A1 emissivity are not available, we deleted these site-days. We have added these details in section 2.1 in the revised manuscript (Line 111-115).

**Relevant text reads (Line 111-115):** "The MOD11A1 surface emissivity has a daily temporal resolution and a 1 km spatial resolution. To obtain the emissivity for each EC flux site, we center on the pixel where the site is located and take the mean value of the 81 neighbouring pixels (9×9 pixels) as the emissivity value of the site. For conditions when the MOD11A1 emissivity are not available, we deleted these site-days."

**R1C6:** Around line 110. The data with negative sensible heat flux with advection (maybe caused by mesoscale circulation or synoptic system) were removed in the research. So maximum evaporation theory can be not used under advections. This is one of difference between relative homogeneous ocean surface and complicated land patches. Please add some discussions on this topic in your discussion part, especially the cautions of applying maximum evaporation theory (limitations?) over saturated

land surface.

**Response:** Done. We removed the negative values for sensible heat to guarantee the data quality. These negative values may be caused by strong advection when accurate measurements are not guaranteed (Paw et al., 2000; Wilson et al., 2002). We have added some discussion and references regarding this point (Line 123-125).

As for the maximum evaporation approach, the basic principles should also hold under the condition of advection except that the Bowen ratio-$T_s$ relationship would be different. Under strong advection conditions, the negative sensible heat may cause an unreasonable Bowen ratio-$T_s$ relationship so the maximum evaporation method may fail. However, out of all available data points, negative $H$ values only account for about 5% of the total daily observations.

**Relevant text reads (Line 123-125):** "Finally, we removed days having a negative $H$ value (account for ~5% of the total daily data) to avoid dealing with strongly advective conditions when accurate measurements are not guaranteed (Paw et al., 2000; Wilson et al., 2002)."

**R1C7:** Around line 125. The calculation of $\tau$ here is same to clearness index. So atmospheric transmissivity here is identical to clearness index?

**Response:** Yes, this reviewer was correct. The shortwave atmospheric transmissivity used here is identical to clearness index.

**R1C8:** Around line 135. "the key processes governing the interactions between incoming and outgoing longwave radiations are essentially the same for ocean and land (mainly greenhouse gas effect)". Firstly, what is the interaction between incoming and outgoing longwave radiation? Secondly, I think the longwave effect process caused by well-mixed GHGs is similar for ocean and land. But clouds and aerosols are different between ocean and land, both two have great effect on longwave radiation.

**Response:** The interaction between incoming and outgoing longwave radiation is that the outgoing longwave radiation would impact the amount of incoming longwave radiation, and vice versa. In our formulation, this interaction is quantified by the temperature difference between the surface and the effective radiating height of the atmosphere.

We agree with this reviewer that besides the GHG effect, aerosols also affect

longwave radiation. In the maximum evaporation approach, the aerosol effect is implicitly considered in the atmospheric transmissivity. We have added the aerosol effect in method (Line 161-162) and disccued it in more details in discussion (Line 294-308) in the revised manuscript.

**Relevant text reads (Line 161-162):** "the key processes governing the interactions between incoming and outgoing longwave radiations are essentially the same for ocean and land (mainly greenhouse gas and aerosol effects)"

**(Line 294-308):** "In the maximum evaporation approach, the coupling between outgoing and incoming longwave radiation is calculated using the temperature difference between the surface and an effective radiating height in the atmosphere ($\Delta T$) and is parameterized as a function of shortwave atmospheric transmissivity and geographic latitude. However, the shortwave atmospheric transmissivity is primarily affected by aerosols while the longwave transmissivity is mainly affected by the concentration of greenhouse gases. Nevertheless, here we only deal with wet conditions, under which the vapour concentration of the atmosphere is also relatively high and more aerosols would favour the development of more clouds that simultaneously affect both shortwave and longwave radiations. We suspect that this underlies the excellent performance of Eq. (5) in estimating $\Delta T$ at the flux sites (Supplementary Figure S2). To further evaluate that conclusion, we additionally evaluate the estimated longwave radiation against four global products (i.e., ERA5, Hersbach et al., 2019; CERES, Kato et al., 2018; the Princeton global forcing data, Sheffield et al., 2006; the GLDAS global forcing data, Rodell et al., 2004) and compare our longwave estimates with other two semi-empirical models (i.e., Brutsaert, 1975 and Shakespeare and Roderick, 2021). The results show our $\Delta T$-based approach to be the best performer across a wide of conditions when the surface is wet (Supplementary Figure S7)."

**R1C9:** Equation (7). You indicated that latent heat of vaporization is a weak function of temperature, so please state this with words and show the calculation formula.

**Response:** Done. Relevent text reads (Line 174-176): "$L$ is the latent heat of vaporization (kJ kg$^{-1}$) and is calculated as weak function of temperature:

$$L(T_s)=2.51\times10^3 - 2.32\times(T_s - 273.15) \tag{9}$$

**R1C10:** Around line 155. You explained why Bowen ratio over land is larger than ocean value in discussion section from stoma resistance. If stoma resistance is the

main reason, Bowen ratio of sparse vegetated land should be close to ocean value, and dense vegetated land should be much higher than ocean value. Can this inference be reflected in Figure2? In addition, aerodynamic resistance for sensible and latent heat flux is thought to decrease with roughness (Zhao et al., 2014). So roughness difference between land and ocean can be used to explain the Bowen ratio difference? Please add some discussion on roughness effect.

**Response:** This reviewer was correct that for a single leaf layer, the stomatal resistance should be higher for dense vegetation than sparse vegetation. However, over densely vegetated land, there are always multiple leaf layers and the stomal resistance for each leaf layer is connected in parallel so the overall canopy resistance is often smaller for dense vegetation than sparse vegetation. As a consequence, the Bowen ratio is usually smaller over densely vegetated lands than over sparsely vegetated lands, when all else is equal. This is also supported by the data showing that croplands and forests have a smaller Bowen ratio than savanna and shrublands for the same surface temperature (Figure 2).

We agree with this reviewer that the roughness difference can be another reason for the Bowen ratio difference between land and ocean. We have added some discussions regarding this point in the revised manuscript (Line 89-91; Line 261-262). Thanks.

**Relevant text reads (Line 89-91):** "In addition, land surfaces usually have a larger surface roughness than ocean surfaces, which may result in a different energy partitioning (into sensible heat and latent heat) between the ocean and the land."

**(Line 261-262):** "In addition, different surface roughness can also lead to different $\beta$-$T_s$ relationships between the land and the ocean."

**R1C11:** "since $T_s$ is very sensitive to changes in $LE$ (Figure 3)" I think it should be "$LE$ is very sensitive to changes in $T_s$" here.

**Response:** It is "$T_s$ is very sensitive to changes in $LE$". As shown in Figure 3, the curve relating $LE$ and $T_s$ is very flat near the maximum evaporation point (where actual evaporation occurs). This means that $LE$ is only a weak function of $T_s$ but a small change in $LE$ can lead to a large change in $T_s$.

**R1C12:** Around line 265. I think the maximum evaporation approach need both incoming solar radiation and reflected solar radiation. If so, using "incoming and reflected solar radiation" is more accurate than "ultimate external forcing".

**Response:** Done. This reviewer was correct. We have revised relevant statement as suggested (Line 79 and Line 338).

**Relevant text reads (Line 79):** "Instead, it only requires the incoming and reflected solar radiation and…"

**(Line 338):** "This new approach only requires the incoming and reflected solar radiation..."

**R1C13:** Symbols and lines are hard to be distinguished in Figure 2. Please improve it. Please add the line of Priestley-Taylor model in Figure 2, which can give some implications for PT model applicability for different land surface.

**Response:** Done. We have adjusted the Figure 2 and made the symbols and lines easier to be distinguished. Comparison of the $\beta$-$T_s$ relationship over wet lands with those for the PT model, the equilibrium evaporation and over ocean surfaces are illustrated in Supplementary Figure S3. Thanks for your suggestion.

[Figure]

**Supplementary Figure S3.** Relationships between the Bowen ratio ($\beta$) and surface temperature ($T_s$).

**R1C14:** "Our results found this held over saturated lands but with considerable scatter (Figure 3)" It should be Figure 2 here.

**Response:** Done. Thanks for pointing out this typo, which has been corrected in the revised manuscript (Line 254).

Reference:

Jasechko, S., Sharp, Z. D., Gibson, J. J., Birks, S. J., Yi, Y., and Fawcett, P. J.: Terrestrial water fluxes dominated by transpiration, Nature, 496, 347-350, https://doi.org/10.1038/nature11983, 2013.

Lian, X., Piao, S., Huntingford, C., Li, Y., Zeng, Z., Wang, X., Ciais, P., McVicar, T., Peng, S., Ottlé, C., Yang, H., Yang, Y., Zhang, Y., and Wang, T.: Partitioning global land evapotranspiration using CMIP5 models constrained by observations. Nature Climate Change, 8, 640-646, 2018.

Maes, W. H., Gentine, P., Verhoest, N. E., and Miralles, D. G.: Potential evaporation at eddy-covariance sites across the globe, Hydrol. Earth. Syst. Sci., 23, 925–948, https://doi.org/10.5194/hess-23-925-2019, 2019.

Paw U, K. T., Baldocchi, D. D., Meyers, T. P., and Wilson, K. B.: Correction of eddy-covariance measurements incorporating both advective effects and density fluxes, Bound-lay. Meteorol., 97, 487-511,    https://doi.org/10.1023/A:1002786702909, 2000.

Wilson, K., Goldstein, A., Falge, E., Aubinet, M., Baldocchi, D., Berbigier, P., et al.: Energy balance closure at FLUXNET sites, Agric. Forest. Meteorol., 113, 223-243, https://doi.org/10.1016/S0168-1923(02)00109-0, 2002.

Yang, Y., and Roderick, M. L.: Radiation, surface temperature and evaporation over wet surfaces. Q. J. R. Meteorol. Soc., 145(720), 1118–1129, https://doi.org/10.1002/qj.3481, 2019.

**To reviewer #2**

**R2C1:**This is an interesting paper, which presents a new framework (in the context of continental surfaces) that could, according to the authors, allow to estimate potential evaporation. I find their approach very "elegant", and the results of this study could be important. However, there are important points that need to be clarified.

The approach ("maximum evaporation theory") is in fact not really new, as its most interesting developments have already been described by the authors in a previous paper, focused on evaporation over ocean ("ocean paper" hereafter). As said by the authors themselves, there is no major reason to expect strong differences between ocean and saturated land. Therefore, the main interest and the main novelty of the paper lie in the evaluation of this approach over land, thanks to a comparison with data from FLUXNET.

It is very difficult to understand the methodology in this paper correctly without carefully reading the ocean paper at the same time, as the authors don't properly justify and discuss the theoretical framework, the assumptions behind their approach, in the submitted paper. They often cite many papers to support their assumptions, but often many of them are not immediately relevant, and the best option for the reader is clearly to directly go to the "ocean paper".

Without explaining everything again in this paper, I think the paper would be much nicer and easier to understand if the authors better explained and justified the main assumptions, limitations etc. of their approach in this paper. It can be done concisely and, in any case, it should not be an issue as the paper is very short (it seems to have been written as a letter). I also think that a few additional analyses should be done. Additionally, important points need to be clarified (see below).

I therefore think that major revisions are needed before the paper could be published.

**Response:** Thanks for your positive evaluation and constructive comments. Following your suggestion, we have added a new section titled "Overview of the maximum evaporation model" to help the readers better understand the approach. In addition, we have clarified the main assumption and uncertainties of the maximum evaporation model in the revised manuscript. Your individual comments are replied below.

**R2C2:** The new method to calculate potential evaporation proposed by the authors in this paper lies on several strong assumptions, not always well justified.

First, the authors hypothesize that "the Bowen ratio is a decreasing function of temperature". The authors cite some theoretical studies that make that point (sometimes indirectly and not very clearly). But I'm quite confused as, as noted in the discussion by the authors themselves, there is a major spread in the observed relationship between the Bowen ratio and Ts (Figure 2). The fit proposed by the authors is quite poor and the explained variance is small.

One could say that based on data shown by the authors, the Bowen ratio is in fact quite poorly controlled by Ts, while in the approach proposed by the authors the Bowen ratio is supposed to be a simple function of Ts.

It seems that either the theoretical arguments are wrong, or H and LE estimates and therefore Bowen ratio estimates from FLUXNET are far from accurate. The authors somewhat acknowledge the issue I stress here in the discussion section, but they seem quite embarrassed by it and to not really know how to deal with it: they don't provide a real conclusion to the discussion of this issue. This should be improved.

**Response:** The decreasing of Bowen ratio with surface temperature under wet conditions has long been tested and validated in numerous previous studies (Andreas et al., 2013, their Figure 1 and Figures 4-6; Guo et al., 2015, their Eq. 4; Philip, 1987, his Figure 1; Priestley and Taylor, 1972, green curve in Figure R3 below; Slatyer and McIlroy, 1961, blue curve in Figure R3 below; Yang and Roderick, 2019, black curve in Figure R3 below). This is also the basis of many other energy balance-based evaporation models, such as the Priestley-Taylor model and the equilibrium evaporation model (see Figure R3 below). This figure is included in the manuscript as Supplementary Figure S3.

[Figure]

**Figure R3.** Relationships between the Bowen ratio ($\beta$) and surface temperature ($T_s$).

According to its definition, the Bowen ratio of equilibrium evaporation ($\beta_e$) can be written as,

$$\beta_e = \gamma \frac{T_s - T_a}{e_s(T_s) - e_s(T_a)} = \gamma \frac{\partial T}{\partial e_s} = \frac{\gamma}{\Delta}$$

where $\gamma$ is the psychrometric constant, $T$ and $e_s$ are temperature and saturated vapor pressure and subscripts $s$ and $a$ stand for surface and near-surface atmosphere, respectively. $\Delta$ is the slope of the saturation vapor pressure – temperature relationship. Since $\gamma$ is a very weak function of temperature and $\Delta$ increases with temperature, so the ratio $\gamma/\Delta$ decreases with temperature. A subsequent study by Priestley and Taylor accounted the fact that the real atmosphere is generally not saturated and modified $\beta_e$ as $\beta_{PT} = 0.79\gamma/\Delta - 0.21$. In our ocean paper, we fitted a Bowen ratio as $\beta_{ocean} = 0.24\gamma/\Delta$ and here we find that $\beta_{wet\ land} = 0.27\gamma/\Delta$. The difference between ocean and wet lands is mainly caused by stomatal resistance of vegetation over land as well as different surface roughness between ocean and land. This difference is discussed in the manuscript (Line 255-263).

The spread of the data points can be caused by many reasons. First, the observations by EC towers can be a source of uncertainty. This is three-fold: (1) the quality of the observations, (2) the footprint within each EC tower may be heterogeneous and (3) whether the selected days are truly non-water-limited still contains uncertainties (however, please see our reply to R2C9). Second, as is seen in our Figure 2, different biome types exhibit different $\beta - T_s$ relationships. This can be caused by different surface resistance and roughness between biome types and even between sites. However, we are not able to parameterize $\beta$ for individual sites due to data limitation. Nevertheless, this limitation only has limited impacts on the model performance, as similar performance is obtained using both the generic $\beta - T_s$ relationship (i.e., $\beta = 0.27\gamma/\Delta$) and biome-specific $\beta - T_s$ relationships (Figure 4). Third, wind speed could be another factor that leads to the spread of the data points. For the same surface roughness, different wind speed lead to different aerodynamic resistance and therefore different Bowen ratio. However, this effect is usually very small, as demonstrated by the long-standing similarity theory (the transfer of mass and heat share the same aerodynamic process in the lower atmospheric boundary layer).

Despite all these effects, we do not intend to incorporate all of them in the calculation of $\beta$ to retain the simplicity (and so the practical application) of the method. On the other hand, incorporating all other effects (or a better model of estimating $\beta$) would not materially affect the model performance, as the sensible heat is generally very small over saturated surfaces.

We have improved the discussion about the data scattering in the revised manuscript (Line 264-277). Thanks for your suggestion.

**Relevant text reads (Line 264-277):** "This data scatter could be caused by several reasons. First, the observations by eddy covariance (EC) towers can be a source of uncertainty. This is threefold, including (i) the quality of the observations, (ii) the footprint within each EC tower may be heterogeneous (Lee et al., 2004; Paw et al., 2000), and (iii) whether the selected days are truly non-water-limited (however, see Supplementary Figure S4). Second, as is seen in Figure 2, different biome types exhibit different $\beta$-$T_s$ relationships. This can be caused by different surface resistance and roughness between biome types and even between sites. Nevertheless, these data-based limitations only have limited impacts on the model performance, as similar performance is obtained using both the generic $\beta$-$T_s$ relationship (i.e., $\beta = 0.27\gamma/\Delta$) and biome-specific $\beta$-$T_s$ relationships (Figure 4). Third, wind speed could be another factor that leads to the scatter. For the same surface roughness, a different wind speed will lead to a different aerodynamic resistance and therefore a different $\beta$. However, this effect is usually very small, as demonstrated by the long-standing similarity theory (the transfer of mass and heat share the same aerodynamic process in the lower atmospheric boundary layer; Monin and Obukhov, 1954)."

**R2C3:** Second, if we accept the assumptions made in the paper, I agree that there exists a maximum evaporation along the Ts gradient. However, I don't understand why the actual evaporation should be equal to this maximum evaporation given by their model. An infinity of pairs of (evaporation, Ts) values are compatible with the authors' model. The authors do not discuss this point at all. Maybe I am missing something obvious.

I agree that the analysis of observations suggests that the maximum evaporation calculated with the authors' approach is close to the observed evaporation (when there is no water limitation) but could the authors justify, based on physical arguments, why the actual evaporation should be equal to the maximum evaporation given by their model?

**Response:** We do not understand the comment, "An infinity of pairs of (evaporation, $T_s$) values are compatible with the authors' model.", as there is only one maximum evaporation and one corresponding $T_s$ along the entire $T_s$ range. More importantly, we did not invoke any maximization (or minimization) assumption in the development of the method, the maximum evaporation emerges naturally from the trade-off between decreased net radiation and increased evaporative fraction as $T_s$ increases. Compared

with observations (over both ocean and wet land surface), this maximum evaporation corresponds to actual evaporation and the $T_s$ at which the maximum evaporation occurs also corresponds to the observed $T_s$. This means that the method correctly captures the interactions between radiation, surface temperature and evaporation. This also explains why the maximum evaporation corresponds to the actual evaporation, because the method simultaneously recovers the observed $T_s$. We believe that this reviewer would accept this more easily if we used the observed $T_s$ to locate evaporation on the evaporation – $T_s$ curve (that will be the maximum evaporation or somewhere near the maximum point). The fact that we do not rely on observed $T_s$ again demonstrates the intrinsic interdependence between radiation, surface temperature and evaporation is correctly captured by the method. Our results also suggest that the maximum evaporation is a natural attribute of saturated surfaces, which results from the trade-off between decreased net radiation and increased evaporative fraction with the increase of $T_s$, as explicitly shown in Yang and Roderick (2019) and in the current study. Following your suggestion, we have added a new section titled "Overview of the maximum evaporation model" in the revised manuscript to help the readers better understand the approach (Line 128-145).

**Relevant text reads (Line 128-145):**

"2.2.1 Overview of the maximum evaporation model

The maximum evaporation model calculates evaporation from a wet surface based essentially on surface energy balance (Eq. (1)) with $R_n$ and $\beta$ both explicitly represented as functions of $T_s$ (Yang and Roderick, 2019):

$$LE = \frac{1}{1+\beta(T_s)}[R_n(T_s)-G] \qquad (3)$$

In the above equation, the first term on the right-hand side (i.e., $1/[1+\beta(T_s)]$) is the evaporative fraction, which is the ratio of the latent heat flux over the total available energy. Over wet surfaces, since the Bowen ratio decreases with $T_s$ (Aminzadeh et al., 2016; Andreas et al., 2013; Guo et al., 2015; Philip, 1987; Priestley and Taylor, 1972; Slatyer and McIlroy, 1961; Yang and Roderick, 2019), evaporative fraction increases with $T_s$. On the other hand, the second term on the right-hand side of Eq. (3) is the total available energy, which decreases with the increase of $T_s$ as a higher $T_s$ directly leads to a higher outgoing longwave radiation and hence a lower $R_n$ (Yang and Roderick, 2019). As a result, the trade-off between a higher evaporative fraction and a lower $R_n$ with the increase of $T_s$ would naturally lead to a maximum $LE$ along the $T_s$ gradient according to Eq. (3). A previous study by Yang and Roderick (2019) have demonstrated that this naturally emergent maximum $LE$ corresponds well to the actual $LE$ over global ocean surfaces and the $T_s$ at which the maximum $LE$ occurs also corresponds to the observed

sea surface temperature. Here we will test whether this maximum evaporation approach is also valid over land under non-water-stressed conditions."

**R2C4:** Third, Delta T in equation (4), and therefore net longwave radiation at surface, is computed thanks to the atmospheric transmissivity for shortwave radiation. It is a huge assumption and it should be discussed.

For example, I don't see how this approach can deal correctly with the impact of aerosols or greenhouse gas (the former having generally an effect on shortwave radiation but not on longwave and conversely for the later). Their approach cannot deal with climate change, right? It should be said. Even for clouds, this assumption is problematic, as some clouds have a strong impact on shortwave radiation, but a weak one on longwave radiation, and conversely.

The authors should discuss this assumption and its limits, and demonstrate that it is reasonable, over land, that they can recover correctly net longwave radiation at surface in a wide range of conditions based on this approach etc.

**Response:** As suggested by this reviewer, we evaluate the estimates of longwave radiation against observations and other global products, and also compared our estimates with other two semi-empirical models. The overall conclusion is that the method used is able to capture net longwave radiation at the surface reasonably well and similarly (or even slightly better) with the other two semi-empirical models across a wide of conditions **when the surface is wet**.

Specifically, Figure R4 (this is Figure 6 in the manuscript) below shows a comparison of estimated net radiation with observed net radiation at the flux sites (across all selected site-days under wet conditions). Since we adopt observed net shortwave radiation, this comparison is essentially the validation of estimated net longwave radiation. It shows that the estimated net radiation corresponds well to the observed ones.

Figure R5 shows a comparison of three models in estimating monthly incoming longwave radiation against global products under wet conditions across the globe (the wet conditions are determined following Milly and Dunne, 2016). The three models include (i) the one used in our study (maximum evaporation model), (ii) the Brutsaert model (1975) and the (iii) Shakespeare-Roderick model (2021). Four global radiation products are used, including (i) ERA5, (ii) CERES, (iii) the Princeton forcing and (iv) the GLDAS forcing. We evaluate incoming longwave radiation here for two reasons: (i) some of the global products do not contain outgoing longwave radiation, and (ii)

the outgoing longwave radiation is estimated based on the Stefan–Boltzmann law, so the real concern lies in the estimation of incoming longwave radiation. Our results show that the maximum evaporation model performs well in estimating incoming longwave radiation across global terrestrial environments when the surface is wet, with a typical RMSE of 20 W m$^{-2}$ and a typical mean bias within ±5 W m$^{-2}$. Compared with the other two methods, the longwave formulation embedded within the maximum evaporation model performs similarly in estimating incoming longwave radiation in terms of RMSE and better than the other two methods in terms of mean bias (Figure R5).

[Figure]

**Figure R4.** Comparison of estimated net radiation ($R_{n\_max}$) with flux site observations ($R_{n\_obs}$).

[Figure]

**Figure R5.** Comparison of model performance in estimating incoming longwave radiation validated against four global products. The three compared models include the maximum evaporation model in this study, the Brutsaert model (1975) and the Shakespeare and Roderick model (2021). The four global products include ERA5 (1979-2019; Hersbach et al., 2019), CERES (2001-2016; Kato et al., 2018), the Princeton global forcing dataset (PGF, 1979-2010; Sheffield et al., 2006) and the GLDAS global forcing dataset (1979-2014; Rodell et al., 2004).

We agree with this reviewer that the greenhouse gases and aerosols impact on shortwave and longwave differently. On the basis of a simple formula for practical applications, our justification for this overall good model performance is that we only

deal with wet conditions. When the surface is wet, relative humidity of the atmosphere is also relatively high. When the atmospheric moisture is sufficient, more aerosols tend to favor the development of more clouds that simultaneously affect both shortwave and longwave radiation. This is different from the conditions such as high aerosol concentrations in dry environments (e.g., deserts), under which the method used herein may fail. However, this is beyond the scope of this study. We have discussed more regarding the uncertainty in the parameterization of the longwave coupling (Line 294-316) in the revised manuscript and included Figure R5 as supplementary material Figure S7 in the revised manuscript to support the validity of our approach.

**Relevant text reads (Line 294-316):** "In the maximum evaporation approach, the coupling between outgoing and incoming longwave radiation is calculated using the temperature difference between the surface and an effective radiating height in the atmosphere ($\Delta T$) and is parameterized as a function of shortwave atmospheric transmissivity and geographic latitude. However, the shortwave atmospheric transmissivity is primarily affected by aerosols while the longwave transmissivity is mainly affected by the concentration of greenhouse gases. Nevertheless, here we only deal with wet conditions, under which the vapour concentration of the atmosphere is also relatively high and more aerosols would favour the development of more clouds that simultaneously affect both shortwave and longwave radiations. We suspect that this underlies the excellent performance of Eq. (5) in estimating $\Delta T$ at the flux sites (Supplementary Figure S2). To further evaluate that conclusion, we additionally evaluate the estimated longwave radiation against four global products (i.e., ERA5, Hersbach et al., 2019; CERES, Kato et al., 2018; the Princeton global forcing data, Sheffield et al., 2006; the GLDAS global forcing data, Rodell et al., 2004) and compare our longwave estimates with other two semi-empirical models (i.e., Brutsaert, 1975 and Shakespeare and Roderick, 2021). The results show our $\Delta T$-based approach to be the best performer across a wide of conditions when the surface is wet (Supplementary Figure S7). In addition, we further note that our maximum evaporation model is only tested at the daily time scale (Figures 4-6) and longer (Figure 3). In particular, for time scales shorter than that (e.g., hourly), the diurnal cycle of $E$ can be very different for ocean and land surfaces (Kleidon and Renner, 2017). In addition, the parameterization of the coupling between incoming and outgoing longwave radiation via Eq. (5) requires a time scale that is long enough to allow the surface heat fluxes to be fully redistributed through the atmospheric column (Yang and Roderick, 2019). At sub-daily scales, Eq. (5) is likely invalid because $R_{lo}$ usually exhibits a larger diurnal range than $R_{li}$ during a typical cloudless day (Monteith and Unsworth, 2013). "

As for the concern on climate change, we did test it (but did not and do not plan to include it in the current manuscript) by incorporating the greenhouse gas effect (primarily the $CO_2$ concentration) in the formulation of $\Delta T$. The sensitivity of incoming longwave radiation on $CO_2$ concentration is determined using MODTRAN.

Figure R6 below shows a comparison between the historical climate (1970-1999, solid curve) and the future climate under the A1B scenario (2070-2099, dashed curve). We estimated that compared with the end of last century, averaged over the entire global ocean, *LE* increases by 4.8 W m$^{-2}$ and sea surface temperature increases by 2.3 K by the end of this century. These are very close to the ensemble of climate model projections of 4.6 W m$^{-2}$ and 2.4 K, respectively. We intend to publish those results in future work as they are beyond the scope of the current manuscript.

[Figure]

**Figure R6.** Variations of *LE* (blue), *H* (red) and $R_n$ (green) with surface temperature averaged over global ocean for historical period (1970-1999, solid curve) and future period under the A1B scenario (2070-2099, dashed curve).

**R2C5:** The authors write that a key issue of energy-balance based models of evaporation is that they consider that "Rn is an independent forcing of E". I don't understand precisely what they mean here, and the references that they cite are not clear. The notion of "independent forcing" is not clear to me. Is a forcing not always independent from the variable it forces, by definition? Additionally, in a climate model, where a coupling exists between the land and the atmosphere, this is not an issue, right? Even in an offline model, with observations, the impact of E on Rn is already included in observed Rn, so why it should be an issue to estimate E?

It is like saying that it is not possible to obtain a realistic simulation with an ocean model forced by observed atmospheric fields (including wind), because the wind field is in fact impacted by sea surface temperature. The implicit impact of sea surface temperature on wind is already included in wind forcing, so this is not an issue.

**Response:** We believe that this reviewer correctly understood our statement of "$R_n$ is not an independent forcing of *E*". Exactly as this reviewer understood, we meant that the impact of *E* on $R_n$ is already included in observed $R_n$. The real underlying issue is $T_s$, because $T_s$ is neither independent of $R_n$ nor evaporation. This is not an issue of estimating evaporation, as demonstrated by long-standing validity of the Penman model and the Priestley-Taylor model in estimating evaporation. However, this is an

issue of understanding evaporation (e.g., attributing evaporation changes by using the Penman model and/or the Priestley-Taylor model). For example, in the existing Penman model, $T_s$ is assumed unknown and that was why Penman developed an approximation. However, the Penman model also assumes $\underline{R}_n$ is known (which requires knowledge of $T_s$). However, as we note in our earlier work on oceans, as $T_s$ increases, $R_n$ actually declines which is the correct physics. However, here we clearly show that evaporation does not always increase with temperature; it depends on the competition with $R_n$-$T_s$ interactions. Moreover, we find that evaporation is not sensitive to changes in $T_s$ but instead, $T_s$ is very sensitive to changes in evaporation. This somewhat suggests that for a given solar radiation, temperature is more of a response rather than forcing of evaporation over wet surfaces.

Correctly understanding and parameterizing these processes/interactions are not only scientific significant but also of important practical uses. For example, here we highlight the implication for estimating potential evaporation, which is the actual evaporation **if the underlying surface were wet**. This implication is beyond the scope of the study but will be addressed in future work. For observations under wet conditions (e.g., the selected wet site-days in the current study), the observed actual evaporation conforms with the definition of potential evaporation, so using observed $R_n$ (or other meteorological forcing required by other models) to estimate potential evaporation is straightforward. However, when the surface is not wet, the observed $R_n$ can be different from the $R_n$ that would have been measured **if the underlying surface were wet** (here we show that $R_n$ decreases with $T_s$, so the observed $R_n$ over a dry surface will be smaller than that if the surface were wet because a wet surface usually has a lower $T_s$ than a dry surface when all else is equal). In the maximum evaporation approach, neither observed $R_n$ nor $T_s$ is required. Our testing results show that the maximum evaporation approach is able to recover the observed $R_n$, $T_s$ and evaporation over wet surfaces indeed suggest the possibility of using this approach to estimate potential evaporation in dry environments. That is why we need the forcing/s to be truly independent. This important implication is discussed in the manuscript (Line 325-344).

**R2C6:** The authors criticize classical approaches to estimate potential evapotranspiration on a theoretical basis, and write that other studies indeed showed that these approaches are not perfect. OK, but their model is also not perfect, some strong assumptions and approximations have to be made, and its results are also not perfect, as shown in the paper. Therefore, they should compare their results with those

obtained with a few common approaches to estimate potential evapotranspiration, using the FLUXNET dataset.

It is not too much work and this analysis clearly should be in this paper, as we want to know whether their model outperforms classical ones. It is possible as the paper is very short.

**Response:** Thanks for the suggestion. Please see our reply to R1C3.

**R2C7:** L75. See my major comments. OK, there is a maximum evaporation, but why this maximum evaporation should be equal to the actual evaporation?

**Response:** Please see our reply to R2C3.

**R2C8:** L83: The authors should discuss how land surface (with no water limitation) and ocean surface differ and how it may impact E.

**Response:** Done. We have added discussion on the difference between ocean and wet land surfaces in the introduction (Line 86-93) to better inform our motivation. Thanks for the suggestion.

**Relevant text reads (Line 86-93):**"Testing the maximum evaporation theory over land is important, as vegetation transpiration generally dominants the total evaporative flux over land (Jasechko et al., 2013; Lian et al., 2018), which is essentially different from ocean surfaces where the evaporative flux only consists of evaporation from open water surfaces. In addition, land surfaces usually have a larger surface roughness than ocean surfaces, which may result in a different energy partitioning (into sensible heat and latent heat) between the ocean and the land. Therefore, it is crucial to test the maximum evaporation theory over land to determine whether saturated land behaves like the ocean surface and whether the maximum evaporation theory can be the basis of a new approach to estimating $E_P$ over land."

**R2C9:** L110: The selection of the days without water limitation seems very ad-hoc and subjective, with for example the step "50% of maximum soil moisture (taken to be the 98$^{th}$ percentile)".

How were the criteria chosen? Trial and error? How can we be sure that the criteria lead to a good separation of days with or without water limitation? Maybe the

separation is not that good, which could be explain why the observed relationship between the Bowen ratio and Ts is not really the one expected by the authors?

More generally, are the results of the paper sensitive to the criteria used to select the days without water limitation? This should be tested.

**Response:** Done. The selection of days without water limitation is largely based on the same selection criteria given by Maes et al. (2019). The 50% of maximum soil moisture is chosen because the field capacity of soil (evaporation is generally not limited by water if the soil moisture were higher than field capacity) usually lies in a range of 33% - 50% of the saturation point (assumed to be the maximum soil moisture at each site). The "98th percentile" is also directly taken from Maes et al. (2019). Although they did not explain why the "98th percentile" was used, we suspect that this is to ensure that the selection is not affected by a few unrealistic high soil moisture records commonly present in the FLUXNET dataset. More importantly, as also pointed out by this reviewer, the model performance is not sensitive to the selection criteria (see Figure R7, where the soil moisture criterion is set to 30% – 70% of the maximum soil moisture, the maximum soil moisture criterion is set to the 95th – 99th percentile, and the evaporative fraction criterion is set to 0.5 – 0.9. We have added some discussion regarding to the uncertainty in the selection of non-water stressed observations (Line 317-324) and included Figure R7 as supplementary Figure S4 in the revised manuscript.

**Relevant text reads (Line 317-324):** "As for the selection of non-water-stressed evaporation observations from global EC towers, we rely largely on the same selection criteria used in a previous study (Maes et al., 2019). However, these selection criteria are somewhat subjective and represent a compromise between better data quality and more data samples. As a result, the selected site-days are not necessarily non-water-limited. Nevertheless, varying the selection criteria (changing the thresholds) of non-water-stressed evaporation only resulted in minor changes in the overall model performance (Supplementary Figure S4), which suggests that the uncertainties in the selection of non-water-stressed evaporation observations would not materially change our conclusion. "

[Figure]

**Figure R7.** Model performance in estimating *LE* with varying selection criteria of unstressed evaporation observations. (a) The soil moisture criterion varies from 30%

to 70%, (b) the Maximum soil moisture criterion varies from 95$^{th}$ to 99$^{th}$ percentile and (c) the evaporative fraction criterion varies from 0.5 to 0.9.

**R2C10:** L112. "To avoid dealing with strongly advective condition we additionally removed days having a negative H value." Are these conditions frequent? It is important to provide this information as if the approach proposed by the authors cannot deal with a large number of days, it limits its real-world applicability to estimate potential evaporation.

**Response:** The removal of strong advection condition is mainly to ensure the data quality as reliable observations by the EC tower is not guaranteed under strong advection conditions (Paw et al., 2000; Wilson et al., 2002). Following your comments, we count the proportion of negative $H$ values in the datasets. Out of all available data points, negative $H$ values only account for about 5% of the total daily observations. We have added some discussion on this point (Line 123-125) in the revised manuscript.

**Relevant text reads (Line 123-125):** "Finally, we removed days having a negative $H$ value (account for ~5% of the total daily data) to avoid dealing with strongly advective conditions when accurate measurements are not guaranteed (Paw et al., 2000; Wilson et al., 2002)."

References:

Aminzadeh, M., Roderick, M. L., and Or, D.: A generalized complementary relationship between actual and potential evaporation defined by a reference surface temperature, Water. Resour. Res., 52, 385–406, https://doi.org/10.1002/2015WR017969, 2016.

Andreas, E. L., Jordan, R. E., Mahrt, L., and Vickers, D.: Estimating the Bowen ratio over the open and ice-covered ocean, J. Geophys. Res. Oceans, 118, 4334–4345, https://doi.org/10.1002/jgrc.20295, 2013.

Brutsaert, W.: On a derivable formula for long-wave radiation from clear skies. Water. Resour. Res., 11(5), 742-744, https://doi.org/10.1029/WR011i005p00742, 1975.

Guo, X., Liu, H., and Yang, K.: On the application of the Priestley–Taylor relation on sub-daily time scales, Boundary. Layer. Meteorol., 156, 489–499, https://doi.org/10.1007/s10546-015-0031-y, 2015.

Hersbach, H., Bell, B., Berrisford, P., Hirahara, S., Horányi, A., Muñoz Sabater, J., et

al.: The ERA5 global reanalysis. Q. J. R. Meteorol. Soc., 146(730), 1999-2049, https://doi.org/10.1002/qj.3803, 2020.

Kato, S., Rose, F.G., Rutan, D.A., Thorsen, T.J., Loeb, N.G., Doelling, D.R., et al.: Surface irradiances of edition 4.0 Clouds and the Earth' s Radiant Energy System (CERES) Energy Balanced and Filled (EBAF) data product. J. Clim., 31, 4501–4527, https://doi.org/10.1175/JCLI-D-17-0208.1, 2018.

Maes, W. H., Gentine, P., Verhoest, N. E., and Miralles, D. G.: Potential evaporation at eddy-covariance sites across the globe, Hydrol. Earth. Syst. Sci., 23, 925–948, https://doi.org/10.5194/hess-23-925-2019, 2019.

Milly, P. C. D., and Dunne, K. A.: Potential evapotranspiration and continental drying, Nat. Clim. Change, 6, 946–949, http://dx.doi.org/10.1038/nclimate3046, 2016.

Paw U, K. T., Baldocchi, D. D., Meyers, T. P., and Wilson, K. B.: Correction of eddy-covariance measurements incorporating both advective effects and density fluxes, Bound-lay. Meteorol., 97, 487-511,    https://doi.org/10.1023/A:1002786702909, 2000.

Philip, J. R.: A physical bound on the Bowen ratio, J. Clim. Appl. Meteorol., 26, 1043–1045, https://doi.org/10.1175/1520-0450(1987)026<1043:APBOTB>2.0.CO;2, 1987.

Priestley, C. H. B., and Taylor, R. J.: On the assessment of surface heat flux and evaporation using large-scale parameters, Mon. Weather. Rev., 100, 81–92, https://doi.org/10.1175/1520-0493(1972)100<0081:OTAOSH>2.3.CO;2, 1972.

Rodell, M., Houser, P. R., Jambor, U. E. A., Gottschalck, J., Mitchell, K., Meng, C. J., et al.: The Global Land Data Assimilation System. Bull. Amer. Meteor. Soc., 85, 381–394, https://doi:10.1175/BAMS-85-3-381, 2004.

Shakespeare, C. J., and Roderick, M. L.: The clear sky downwelling longwave radiation at the surface in current and future climates. Q. J. R. Meteorol. Soc., https://doi.org/10.1002/qj.4176, 2021.

Sheffield, J., Goteti, G., and Wood, E.F.: Development of a 50-year high-resolution global dataset of meteorological forcings for land surface modeling. J. Clim., 19, 3088–3111, https://doi.org/10.1175/JCLI3790.1, 2006.

Slatyer, R.O., and McIlroy, I.C.: Practical microclimatology, Commonwealth Scientific and Industrial Research Organisation, Canberra, Australia, 1961.

Wilson, K., Goldstein, A., Falge, E., Aubinet, M., Baldocchi, D., Berbigier, P., et al.: Energy balance closure at FLUXNET sites, Agric. Forest. Meteorol., 113, 223-243,

https://doi.org/10.1016/S0168-1923(02)00109-0, 2002.

Yang, Y., and Roderick, M. L.: Radiation, surface temperature and evaporation over wet surfaces. Q. J. R. Meteorol. Soc., 145(720), 1118–1129, https://doi.org/10.1002/qj.3481, 2019.

---

## Author Response (AR2)

Response to Reviewers' comments

We greatly appreciate the editor and the two referees for providing constructive comments that are of great help for us to improve the manuscript. We have fully considered the comments and the point-to-point responses are listed below. The manuscript and the supplementary material have been revised to accommodate the changes (changes are marked in blue color). In the following the reviewer comments are black font and our responses are blue and to assist with navigation we use codes, such as R1C2 (Reviewer 1 Comment 2)

**To reviewer #2**

R2C1: Overall, I greatly appreciate how the authors dealt with my comments (and the other reviewer's comments), with clear detailed answers and new analyses when necessary. I also appreciate how they modified the manuscript accordingly.

I think the paper is almost ready for publication, except for the few (but important) comments below, and will be an excellent contribution to the field.

Reply: Thanks for your positive evaluation of our revised manuscript. Your individual comments are replied below.

R2C2: I am not really satisfied by the response to my comment R2C3 in the previous review: My question was: Why is the actual evaporation equal to the maximum evaporation? The authors answered "We do not understand the comment, "An infinity of pairs of (evaporation, Ts) values are compatible with the authors' model.", as there is only one maximum evaporation and one corresponding Ts along the entire Ts range."

I was indeed not clear: I should have said: "An infinity of pairs of (evaporation, Ts) values are compatible with Equation (3) (in the new version of the manuscript)". And Equation (3) is not sufficient to estimate actual evaporation in the authors' framework. To estimate actual evaporation, the authors also have to make the additional assumption that actual evaporation is equal to maximum evaporation.

The model proposed by the authors to estimate evaporation actually needs two "ingredients":

(i) Equation (3) that describes how evaporation evolves with Ts, and shows the existence of a maximum evaporation at a given Ts

AND

(ii) The assumption that actual evaporation is equal to this maximum evaporation

To my question: "Why is the actual evaporation equal to the maximum evaporation?", basically the authors answered: if we look at observations, it works. I agree, but this does not explain why.

Their model is therefore empirical to an important extent, which is somewhat less satisfying from an intellectual point of view. It should be acknowledged.

It is not that important for the model proposed by the authors in practice, as it is true that "it works" according to observations, but I think it should be made clear that the model relies both on equation (3) and on assumption (ii) above, and that assumption (ii) is only justified empirically in this work.

Reply: We now understand this point better. However, we did not invoke the assumption that the maximum evaporation is equal to actual evaporation in the method. We did not know whether and to what extent the maximum evaporation corresponds to actual evaporation. In fact, we did not even know what this maximum evaporation means at the beginning. The maximum evaporation emerges naturally from the intrinsic interactions between radiation, surface temperature and evaporation. The fact that this maximum evaporation corresponds to the actual evaporation indicates that the maximum evaporation is a natural attribute of extensive wet surfaces. So a better summary of our previous study on ocean surfaces is not "developed a new evaporation model" but "identified this natural attribute of global ocean surfaces". In the current study, we further demonstrated that this is also a natural attribute of wet surfaces over land (with vegetation). These findings combined lead to an important conclusion that the maximum evaporation is a natural attribute of an extensive wet surface, regardless of the surface is covered by water, soil or vegetation.

So a clear answer to your question "Why is the actual evaporation equal to the maximum evaporation?" is "the maximum evaporation is a natural attribute of an extensive wet surface".

R2C3: L161-162. "the key processes governing the interactions between incoming and outgoing longwave radiations are essentially the same for ocean and land (mainly greenhouse gas and aerosol effects)".

It is not true: the influence of most aerosols on longwave radiation is very weak. And the influence of clouds and of the vertical temperature structure of the atmosphere are crucial for longwave radiation.

Reply: We agree with this reviewer. However, water vapor (the largest greenhouse gas in the atmosphere) and aerosols are the key foundations of clouds, and the concentration of greenhouse gases also largely determines the vertical temperature structure in the atmosphere. To avoid potential misunderstandings, we have clarified this point by revising the statement as "mainly greenhouse gases that affect the vertical temperature structure of the atmosphere, and water vapor and aerosols that affect the formation of clouds) (Line 162-164).

R2C4: L294-308. "However, the shortwave atmospheric transmissivity is primarily affected by aerosols while the longwave transmissivity is mainly affected by the concentration of greenhouse gases"

And what about clouds? Generally, the impact of aerosols on shortwave radiation is not greater than the one of clouds. And clouds may have very different impacts with regards to the shortwave and longwave radiation budgets. For example, high clouds have a strong impact on longwave radiation compared to low clouds, but the same cannot be said for shortwave radiation.

Reply: We agree with this reviewer and the different impacts of clouds on shortwave and longwave are accounted for in the scaling relationship between $\Delta T$ and $\tau$.

R2C5: L310. Thanks for the very detailed answer to my previous comment on this point: it is very interesting.

I still think it is important to say in the paper that there may be some issues at longer time-scales, in the climate change context, as the relationship between shortwave transmitivity and longwave radiation is expected to evolve with anthropogenic climate change.

Reply: Done. We have added a sentence that reads "For even longer periods, especially for assessing the impacts of climate change, the relationship between shortwave and longwave radiations used herein may be also invalid, as we expect this relationship to evolve with anthropogenic climate change." (Line 317-319). Thanks for the suggestion.

**To reviewer #3**

R3C1: With great interest I read the revised manuscript and the author's response. The authors apply their previously found model to estimate 'maximum evaporation' over ocean to saturated land evaporation. The proposed method is in my view very elegant due to its simplicity. Equation 3 is the core equation from where it can be easily seen that Ts has positive and negative feedbacks on LE (first term positive, 2nd one negatively correlated to LE). In that sense I think that maybe 'equilibrium evaporation' might be a better term than 'maximum evaporation', although I can see from an optimisation point of view that it's indeed the max. Using the term equilibrium might also help to understand why LEmax corresponds to actual evaporation under saturated conditions.

Reply: Thanks for your encouraging comments on our revised manuscript. However, we feel that "equilibrium evaporation" is not an appropriate term, as there has already been a specific definition of "equilibrium evaporation" – evaporation from a saturated surface into a saturated atmosphere (Schmidt, 1915; Slatyer and McIlroy, 1961; Philip, 1987). This is different from our maximum evaporation, where we do not assume a saturated atmosphere. The maximum evaporation emerges naturally from the intrinsic interactions between radiation, surface temperature and evaporation. The fact that this maximum evaporation corresponds to the actual evaporation indicates that the maximum evaporation is a natural attribute of extensive wet surfaces. Therefore, we use "maximum evaporation" to highlight this is a natural attribute of extensive wet surfaces.

R3C2: Only surprising point for me remains the low sensitivity of all energy balance fluxes to Ts (Figure 3), especially, in the domain of interest: 280<Ts<310 (although already quite elaborately discussed in the manuscript and author-response: low sensitivity of the bowen ratio for Ts more most observation points). The results for LE and Rnet are excellent (figure 4 and 6), those for Ts are moderate/good (figure 5). Leaving a comparison on H open. How are these results? Might it be possible that all 'errors' are allocated to H? Also partly related to the way how you dealt with the energy balance gap in your EC-data.

Reply: The low sensitivity of evaporation to Ts is actually somewhat expected and implied in many previous studies. This is because that the incoming longwave radiation increases concurrently with the outgoing longwave radiation (due to the increase of Ts), leaving changes in net radiation very small. This also partly explains the overall low hydrological sensitivity to climate change (1.5% - 2% increases in the strength of the global hydrological cycle with a 1°C increase of temperature). As for

the sensible heat, it is much more difficult to correctly capture the sensible heat than for the latent heat, because under saturated conditions, the sensible heat is usually very small. Following your suggestion, we had a quick look at the sensible heat estimates and found an overall RMSE of 8.3 ~ 9.7 W m$^{-2}$ and a mean bias of 0 ~ 0.8 W m$^{-2}$ (the range is due to using different methods to close the EC-tower energy balance and using the generic or biome-specific β-Ts relationships). This is not bad at all.

R3C3: Nonetheless, I highly appreciated to read this manuscript and looking forward to the next step: how does it work under unsaturated conditions?

Reply: Very good point. We are actually working on this.

Reference:

Schmidt, W. (1915), Strdhlung und Verdunstung an freien WasserflAchen; ein Beitrag zum Warmehaushalt des Weltmeers und zum Wasserhaushalt der Erde (Radiation and evaporation over open water surfaces; a contribution to the heat budget of the world ocean and to the water budget of the earth), Annalen der Hygrographir und Maritimen Meteorologie, 43, 111-124.

Slatyer, R.O. and McIlroy, I.C. (1961) Practical Microclimatology. Melbourne: Commonwealth Scientific and Industrial Research Organisation.

Philip, J.R. (1987) A physical bound on the Bowen ratio. Journal of Climate and Applied Meteorology, 26, 1043–1045.